# Gene Therapy in Cancer Treatment: Why Go Nano?

**DOI:** 10.3390/pharmaceutics12030233

**Published:** 2020-03-05

**Authors:** Catarina Roma-Rodrigues, Lorenzo Rivas-García, Pedro V. Baptista, Alexandra R. Fernandes

**Affiliations:** 1UCIBIO, Departamento de Ciências da Vida, Faculdade de Ciências e Tecnologia, Campus de Caparica, 2829-516 Caparica, Portugal; catromar@fct.unl.pt (C.R.-R.); lorenrivas@ugr.es (L.R.-G.); 2Biomedical Research Centre, Institute of Nutrition and Food Technology, Department of Physiology, Faculty of Pharmacy, University of Granada, Avda. del Conocimiento s/n. 18071 Armilla, Granada, Spain

**Keywords:** gene therapy, gene delivery, tumor microenvironment, nanoparticles, nanomedicine

## Abstract

The proposal of gene therapy to tackle cancer development has been instrumental for the development of novel approaches and strategies to fight this disease, but the efficacy of the proposed strategies has still fallen short of delivering the full potential of gene therapy in the clinic. Despite the plethora of gene modulation approaches, e.g., gene silencing, antisense therapy, RNA interference, gene and genome editing, finding a way to efficiently deliver these effectors to the desired cell and tissue has been a challenge. Nanomedicine has put forward several innovative platforms to overcome this obstacle. Most of these platforms rely on the application of nanoscale structures, with particular focus on nanoparticles. Herein, we review the current trends on the use of nanoparticles designed for cancer gene therapy, including inorganic, organic, or biological (e.g., exosomes) variants, in clinical development and their progress towards clinical applications.

## 1. Introduction

According to the World Health Organization, cancer is the second leading cause of death worldwide, accounting for 9.6 million deaths in 2018 [1]. The global efforts in cancer prevention, early diagnosis, screening and treatment, have been challenged by the complexity and variability of tumors (reviewed in [2]). The genomic instability of tumor cells and a pro-inflammatory environment are key factors for tumor growth [3]. Regardless of the monoclonal origin of the neoplasia, the interplay between tumor cells and the surrounding environment results in a complex tumor microenvironment (TME) that supports tumor intra-heterogeneity, with spatially different and phenotypically distinct subclones [2]. Nonetheless, major common features of tumor cells include continuous proliferative signaling, evasion of growth suppressors, resisting cell death, replicative immortality, deregulating cellular energetics, promoting angiogenesis, activating invasion and metastasis, and avoiding immune destruction [3]. These features sustain the foundation of a TME composed by a characteristic extracellular matrix (ECM), cancer-associated fibroblasts (CAFs), mesenchymal stromal cells, endothelial cells and pericytes, and immune system cells, such as macrophages, T and B lymphocytes and natural killer cells (reviewed in [4]). TME composition dictates tumor progression, chemotherapeutic efficacy and prognosis [5,6,7].

The mounting knowledge on the characteristics of tumor cells and surrounding TME have sparked the use of gene therapy to tackle cancer molecular mechanisms. Gene therapy consists of the introduction of exogenous nucleic acids, such as genes, gene segments, oligonucleotides, miRNAs or siRNAs into cells envisaging a target gene edition, expression modulation of a target gene, mRNA or synthesis of an exogenous protein [8,9,10,11,12,13,14,15,16,17,18,19]. Gene transfer into tumor cells has been demonstrat*ed* via administration of therapeutic nucleic acids (TNAs) ex vivo and/or in vivo (Figure 1) [20]. In the ex vivo approach, patient-derived tumor cells are collected, propagated usually as 2D monolayers, manipulated genetically and then introduced back into the host [20]. In the in vivo approach, TNAs may be introduced *in loco* into the tumor cells, systemically via intravenous administration, or in a pre-systemic manner through oral, ocular, transdermal or nasal delivery routes, depending of the specific localization of tumors and disease progression [20,21,22]. In the case of systemic and pre-systemic deliveries, the administration of naked TNAs is hindered by biological barriers, nuclease susceptibility, phagocyte uptake, renal clearance and/or immune response stimulation [23]. Hence, the use of stable carriers/vectors that protect the nucleic acid cargo from circulatory nucleases, avoid the immune system, and ensure the efficient targeting of the therapeutic vector into the tumor cells, without dissipation in the body through lymphatic and blood systems and avoiding non-target cells is required [21]. Despite the apparent limitations of the *in vivo* approach, it is less invasive and more suitable for cancer treatment than ex vivo approaches, since the latter require a proliferative advantage of transfected cells, which is antagonist to the major objectives of cancer gene therapeutics that mainly aims to inhibit the tumor progression by tackling the tumor cell division ability [21,24,25,26]. Nevertheless, it is important to highlight the relevance of ex vivo therapy in indirect immune gene-based therapies (detailed in Section 2.7). In these *ex-vivo* approaches, immune cells are collected from the patient’s blood and genetically engineered to tackle the tumor cells (reviewed in [27,28]).

The success of cancer gene therapy relies on a safe, effective and controllable vector [25]. Viral vectors were the first platform proposed for gene therapy [29]. Indeed, the nature and properties of viruses made them tempting vectors for RNA and DNA delivery to human cells, with multiple clinical trials that ended-up in clinical approved of some gene therapy drugs (reviewed in [29]). However, the immunogenicity, limited genetic-load, cancer risk due to therapeutic payload insertion near genes that control cell growth, and constrained mass-production of viral vectors prompted the development and engineering of non-viral vectors, supported by nanomedicine [25,30].

Nanotechnology refers to the area of science focused on the study of the synthesis, characterization and application of materials and functional systems of particles whose size is between 1-100 nm [31,32]. Nowadays, the interest in these materials is not only due to their small size, but also to their unique physical (electric, optical, magnetic) and chemical properties at these dimensions (in comparison to the same material at the macroscopic scale), conveying a more scalable interaction with cells and biomolecules. The application of nanotechnology to the medical field (nanomedicine) enhanced the development of new and more effective diagnostics and therapies, particularly in complex diseases, such as diabetes, Parkinson and cancer [33,34]. The application of nanoparticles as carriers in gene therapy is one of the most promising technologies in biomedical research due to its ease and straightforward synthesis and functionalization with different moieties, low immunogenicity and toxicity [35]. One of the interests is focused on the development of biocompatible and more effective transfection systems [36,37] to vectorize TNAs to cells and tissues, such as DNA (e.g., plasmid DNA, antisense oligonucleotides (ASO)) or RNA (e.g., microRNA (miRNA), short hairpin RNA (shRNA), small interfering RNA (siRNA)) into cells [36,37]. Some of the limitations in the efficiency of transfection of naked plasmid DNA (pDNA) or siRNAs can be improved by the application of functionalized nanoparticles [38]. However, this technology still faces several shortcomings that need to be addressed, including lower transfection efficiency when compared to viral vectors, or blood clearance before reaching the target site in the case of systemic administration [39].

Effective TNA agents require a vector that can travel through the circulatory system, accumulate in the tumor, enter target cells via endosome pathway and be able to escape the endosome to efficiently accomplish cargo delivery (Figure 2) [21,23,25].

Indeed, systemic administration of the therapy implies that the vector can travel through the blood circulation, with consequent interaction with blood cells, including phagocytic cells, proteins and lipids [40]. The surface, size and shape of the nanoparticles are preponderant for their endurance across the circulatory system (reviewed in [40]). Neutral or slightly negative surfaces assure low adsorption to blood proteins, such as opsonins, and avoid phagocytosis (reviewed in [40,41]). Hence, neutralization of charged nanoparticles may be achieved by coating with hydrophilic polymers such as polyethylene glycol (PEG), polyglycerol (PG), or polysaccharides, such as heparan or chitosan, with zwitterionic ligands, such as carboxybetaines or sulfobetaines, with mercaptoalkyl acid ligands, such as 11-mercaptoundecanoic acid (MUA), or even with proteins and lipids (reviewed in [40,42]).

Concerning tumor accumulation, nanomedicine design often takes advantage of the natural accumulation at the tumor location – passive targeting [40,43,44]. In fact, the characteristic immature phenotype of the tumor vasculature, characterized by leaky vessels with chaotic branching, together with poor lymphatic system, renders an enhanced permeability and retention (EPR) of nanoparticles at the TME (reviewed in [43,44,45]). As the EPR effect is dependent of the tumor in terms of the anatomical location, tumor size, stage and type, the properties of the nanoparticles (size, shape and surface charge) should be optimized (reviewed in [40]). As an example, pegylation of drug-loaded liposomes not only improved their blood circulation, but also increased the accumulation of the drug in the tumor [43]. Furthermore, active targeting of the nanomedicines improves greatly their efficacy (reviewed in [40,46,47,48]). With that purpose, several biological ligands could be bind to nanomaterials, including antibodies, such as cetuximab, an FDA approved antibody against anti-epidermal growth factor receptor (EGFR) used in clinical practice for cancer treatment; glycoproteins, such as the iron-binding transferrin; polysaccharides, such as hyaluronic acid for CD44 targeting; peptides, such as arginylglycylaspartic acid (RGD) for integrins targeting; aptamers, such as AS-1411 G-rich DNA aptamer for nucleolin targeting; or other small molecules, such as folate ([49,50,51,52,53,54,55,56,57,58,59,60], reviewed in [48]).

After reaching the tumor, another bottleneck that nanoparticles must overcome is the penetration into the TME to reach regions with low or without vascularization, low interstitial fluid pressure and dense ECM [61,62,63]. To achieve higher diffusion, the nanoparticles size, functionalization and tumor modulation ability have been extensively studied and modulated (reviewed in [63,64]). This is often accomplished by altering the nanoparticles’ properties, such as size and surface hydrophobicity, after they reach the tumor using stimulus-triggering strategies, including light, ultrasound or magnetic fields, or taking advantages of the TME properties, such as hypoxia, acidity, and the overexpression of metalloproteinases (reviewed in [40,63]).

Additionally, for gene therapy it is necessary that the vector and payload pass across the complex hydrophobic layer of the tumor cell membrane [41]. This mainly occurs via endocytosis mediated by ligand-receptor specific, using active targeting, or non-specific, such as electrostatic or hydrophobic, interactions with the cell membrane (reviewed in [40,65,66]). Once more, nanoparticle size, shape and surface are preponderant for an efficient cellular internalization [67,68,69,70,71,72,73]. The most suitable features are dependent on type of particle, for example, Xue et al. observed an improved internalization when polypeptide-based nanoparticles composed by mixtures of FITC-poly(γ-benzyl-L-glutamate)-block-PEG and polystyrene (PS) had smaller size, rod-like shape and helical/striped surface morphology [68]. On the other side, a study from Bandyopadhyay et al. revealed that the pillow-shape and irregular structure of gold nanoparticles resulted in a higher cellular uptake, when compared with small spherical nanoparticles [70]. Once nanoparticles enter cells via endocytosis, they must escape from the endosome to avoid degradation in lysosomes or exocytosis [65]. The endosomal escape could occur through membrane destabilization, proton sponge or photochemical internalization [40,65,74]. Liposomes and lipid-based nanoparticles mainly escape via membrane destabilization due to direct contact between the nanoparticles lipid layer with the endosomal membrane, resulting in the release of the nanoparticles content into cytoplasm [65]. The lipidic nanoparticles transport of membrane-disruptive peptides induce the formation of a pore in the endosome, enhancing the endosomal escape [74]. The proton sponge escape occurs when nanoparticles containing amino groups, such as polyethylenimine (PEI) or polyamidoamine (PAMAM) based dendrimers, are protonated during acidification of the endosome, resulting in an increased osmotic pressure due to an influx of chloride ions and consequent swelling of the endosome with nanoparticles release into the cytoplasm [65,74]. In the photochemical internalization, nanoparticles are functionalized with photosensitizers that after light activation generate reactive oxygen species (ROS) that rupture the endosomal membrane [74]. The most suitable properties of nanoparticles that enhance endocytosis and endosomal escape was extensively reviewed in recent papers [65,74,75].

Considering all the characteristics for an ideal vector towards effective gene therapy and the advantages posed by nanoparticles as gene delivery systems, the present review will first address the different non-viral gene therapy strategies used in cancer, followed by the application of the different nanoparticles as vectors for cancer therapy, together with their way to the clinics.

## 2. Gene Therapy Focused on Cancer

Delivery of TNAs, such as genes, oligonucleotides, miRNAs or siRNAs to cancer cells has allowed to tackle cancer via the silencing oncogenes or restoring the expression of tumor suppressor genes [8,9,10,11,12,13,14,15,16,17,18,19]. Most of these approaches (e.g., antisense therapy, RNA interference (RNAi), gene editing) aim at gene alteration/modulation [16,17,18,19,76,77,78,79,80,81] - see Figure 3. The immunization gene therapies, particularly chimeric antigen receptor (CAR) in T cells (CAR-T cells) based therapies, stand-out since they represent the higher number of therapeutic strategies in clinical practice. It should be noted that some of the presented strategies such as genome editing or miRNA/siRNA targeted therapy are used in TME targeting via angiogenesis and immune therapies [82,83] or CAFs targeting [84,85,86].

### 2.1. Oncogene Silencing via RNAi

Gene silencing consist in the delivery of nucleic acids into tumor cells that end up in downregulation of specific genes [24,37,87,88]. Gene silencing therapy is usually accomplished by introducing siRNA or shRNA in tumor cells designed to target a specific complementary sequence to messenger RNA (mRNA) of a selective gene, inducing its degradation or by blocking protein synthesis [89]. Oncogenes, such as *cMYC* or *KRAS,* and genes involved in drug-resistance such as multi-drug resistance 1 (MDR1) are tempting targets for tumor therapy using RNAi [14,24,90,91,92,93,94]. Major challenges faced by RNAi are related to target specificity, off-target RNAi activity, dissipation in circulation, cellular internalization and endosomal escape [95]. Strategies used to surpass these limitations were extensively reviewed in [12,95].

### 2.2. Tumor Suppressor Genes Replacement

Gene replacement can be accomplished by gene transduction, maintenance of stability and full expression of the gene, or by correcting gene mutations into its wild-type form (reviewed in [96]). Tumor suppressor genes, such as *TP53*, *P21* or *PTEN* are major targets for gene replacement therapy [25,26,97,98,99,100,101,102]. Due to the central role of P53 protein in cell cycle regulation, DNA repair, apoptosis, senescence and/or autophagy, *TP53* gene is a major target for gene therapy [100]. The first commercial gene therapy product was gendicine in 2003, a recombinant human P53 adenovirus commercialized by SiBiono Gene Technologies, approved by the Chinese Food and Drug Administration for head and neck squamous cell carcinoma [103]. As the bottlenecks of gene editing are transversal for RNA delivery, DNA delivery should also overcome the barrier of the passage through the nucleus membrane [75]. The entry of nucleic acids into the nucleus occur through channels of the nuclear pore complex (NPC, Figure 2), that allow the passage of linear DNA with maximum 200-300 bp, posing a challenge for the nuclear entry of therapeutic gene expression cassettes with few kilobase pairs (kbp) [75]. Strategies to improve the DNA entry into the nucleus involve nuclear-targeted delivery with nuclear localization signals or inclusion of nucleotide sequences in DNA [75]. As the mentioned strategies require activation of signaling pathways that limit their application for cancer therapeutics, another strategy for gene editing therapeutics take advantage of the nuclear envelope disruption during mitosis, which require the presence of intact foreign DNA near the chromatin [75]. The bottlenecks and strategies for gene editing based on nucleic acid delivery were recently reviewed in [75].

### 2.3. microRNA Targeted Therapy

In cancer, some miRNAs are overexpressed promoting tumor development (oncomirs), and others are downregulated bypassing the inhibitory control over oncogenes, or the control of cell differentiation and apoptosis (tumor suppressor miRNAs) [104]. The miRNA targeted therapy consists in the repositioning of the levels of miRNAs in cells. The silence mechanism of miRNA is similar to the RNAi, however, miRNA are complementary or semi-complementary sequences to the 3′ untranslated region (3′-UTR) of a specific mRNA target or to several mRNAs involved in a particular cellular process [12,19]. The levels of miRNAs altered under pathological conditions could be restored to normal physiological conditions using miRNA-duplexes, to replace the levels of underexpressed miRNAs, or siRNA complementary to the seed sequence of the miRNA of interest [20]. Several studies proposed the reposition of miRNAs envisaging cancer therapy, including for example, by adding the tumor suppressor miRNA Let-7c for prostate cancer treatment, by silencing the oncomirs miR-21 in breast cancer, or by over-expressing miR-143 in colon cancer to overcome oxaliplatin resistance [13,104,105]. The systemic administration of free miRNAs for therapy has been challenged by single stranded or double stranded miRNA degradation in the circulatory system or in the endosome, potential off-target effects, miRNA-mediated toxicity and poor delivery [106,107,108]. The knowledge of the miRNAs metabolic modulation in targeted and non-targeted cells is preponderant to avoid off-target effect that occurs due to partial complementarity with non-targeted transcripts, or by leading to undesired effects by regulation of metabolic processes in non-targeted cells [107]. To circumvent the degradation limitations, miRNA may be modified: miRNA mimics are mainly modified by methylation of the passenger strand and locked nucleic acids (LNA) chemistry is used for modification of anti-miRNA (reviewed in [106]). Another strategy involves the delivery of miRNAs in nanoparticles able to perform endosomal escape, reviewed in Section 3, “Nanoparticles for gene delivery: fostering gene therapy”.

### 2.4. Transcription Factor Decoys

Transcription factor decoys (TFD) are double stranded oligodeoxynucleotides (ODN) designed to inhibit specific regulatory pathways (reviewed in [109]). The TFD-ODNs are short double-stranded DNA molecules with the sequence of a transcription factor of a particular gene, or the consensus DNA recognition motif of the transcription factor, competing with specific binding sites of transcription factors [109]. TFDs designed envisaging cancer therapy include TFDs targeting *NF-KB* for inhibition of metastasis, signal transducer and activator of transcription 3, or *STAT3*, to induce apoptosis and cell cycle arrest in ovarian, glioblastoma, lung and neck cancers (reviewed in [110]). Major challenges for the application of TFD-ODNs in cancer therapy include the design of TFDs, and stability in circulatory system and endosome [110]. The design of TFDs require the exact sequence of the transcription factor binding site, which may be a problem due to the mismatch between the available information in databases, and requiring the performance of rigorous but costly techniques such as chromatin immunoprecipitation, and further confirmation of accurate targeting, usually using reporter genes, like luciferase, and usage of scrambled decoys [110]. Peptide nucleic acid (PNA), LNA or phosphorotioate (PS) chemical modifications of the TFD-ODNs could improve their half-life, increase resistance to serum nucleases and decrease the interaction with DNA binding proteins [110]. Nevertheless, the most promising techniques for in vivo TFD-ODN delivery involves their nanoparticle transport [110].

### 2.5. Genome Editing

Genome editing therapy consists in the modification of intracellular DNA in a sequence specific manner, by insertion, deletion, integration or sequence substitution [111,112]. Three major nucleases have been used for this purpose, zinc finger nucleases (ZFN), transcription activator-like effector nucleases (TALEN), meganucleases, and CRISPR/Cas9 system (reviewed in [111]). The efficiency of the genome editing therapy depend on the specificity of the DNA cleavage together with the prevention of collateral damage to the rest of the genome. The CRISPR/Cas9 system was proven as a suitable tool for stable and efficient genome editing as well as for high-throughput screening of mutations involved in oncogenesis and tumor progression [112,113,114,115]. The mostly used CRISPR/Cas9 system is the CRISPR system of *Streptoccocus pyogenes* (SpCas9), that recognize the short sequence 5′-NGG, where N represents any nucleotide and G represents guanine, and Cas9 is an nuclease guided by a single guide RNA (sgRNA) mediated by paring to the target sequence (reviewed in [116]). The CRISPR/Cas9 system is delivered as plasmid or linear DNA encoding Cas9 and sgRNA [116,117]. When delivered as linear DNA, it must enter the nucleus for transcription, while the plasmid DNA allow a stable and prolonged gene expression [116]. The challenges in the delivery of the CRISPR/Cas9 system are similar to other gene editing strategies detailed in Section 2.2. “Tumor suppressor genes replacement”. Moreover, challenges faced by genome editing based on this system are due to prolonged exposure of the genome to endonuclease activity that result in the cleavage of off-target sites [116,117]. The expression of CRISPR/Cas9 system in non-target tissues should be minimized to avoid off-target mutagenesis [116]. The approaches currently used for in vivo delivery of CRISPR/cas9 system were recently reviewed in [116,117].

### 2.6. Suicide Genes

The concept of suicide gene therapy was originally proposed for cancer treatment. Consist in inserting in tumor cells a gene encoding a cytotoxic protein by applying two main strategies: (i) direct gene therapy, by introducing in tumor cells a toxin gene that reduce the viability of the cells, (ii) indirect gene therapy, by introducing a gene encoding an enzyme into tumor cells that is able to convert a non-toxic prodrug into a cytotoxic drug [25,118,119]. The first proposal of suicide gene therapy was made in 1983 by inserting in BALB/c murine cell lines the herpes virus thymidine kinase gene, and then generate tumors with these cells in BALB/c mice [120,121]. Ganciclovir (9-([2-hydroxymethyl)ethoxy]methyl)guanine) was then administered to the mice, and metabolization of ganciclovir by herpes virus thymidine kinase at the tumor cells resulted in tumor regression [120,121]. The potential of this therapeutic strategy motivated its application in several clinical trials for treatment, e.g., liver (NCT02202564) or colorectal (NCT00012155) cancer. The issues inherent to suicide gene therapy are related to gene editing that must result in tumor-specific high expression of the gene, preferentially under control of tumor-specific promoters (reviewed in [122]).

### 2.7. Immunization Gene Therapy

The immunization gene therapy consists in the enhancement of the immune system efficacy towards TME cells, with major focus on tumor cells. Three major approaches are applied, cytokine gene therapy, tumor vaccine therapy and CAR-T cells therapy.

#### 2.7.1. Tumor Vaccines

Tumor vaccination relies on presenting tumor-related antigens to the immune system, triggering an immunological response against cancer antigens/markers (reviewed in [119]). Tumor-related antigens may consist of proteins over-expressed in cancer cells, such as prostate-specific antigen (PSA), differentiation antigens, such as glycoprotein 100, or tumor-specific epitopes [120,121]. The genomic instability in tumor cells result in an alteration of proteins sequence creating new epitopes specific to the tumor, neoepitopes, that can be recognized by T cells [122]. The advent of Next Generation Sequencing (NGS) allowed to obtain a comprehensive mapping of the mutations at the in a specific tumor and prediction of neoepitopes for personalized cancer therapy [122]. This can be accomplished by vaccination the patient with neoepitopes to stimulate the adaptive immune system against tumor cells [25,122]. Vectors for neoepitope presentation include synthetic peptides, mRNA, pDNA, viral vectors, engineered attenuated bacterial vectors or genetically modified APCs, including dendritic cells (DCs), macrophages and activated B cells [122]. DCs showed to be the most promising vaccination vectors, with one DCs-based vaccine approved by the FDA, Sipuleucel-T (Provenge, Dendreon Corporation), for treatment of castrate resistant prostate cancer [123]. However, the tumor point mutations complexity poses limitations for the identification of a neoepitope that will elicit an effective immune response. This subject was recently reviewed in [124,125].

#### 2.7.2. CAR-T Cells Therapy

CAR-T cells therapy is in line with the strategy used in tumor vaccine therapy. In this approach, T cells retrieved from a patient or a healthy donor are genetically engineered to produce antigens against neoepitopes and then are transferred back to the patient [126]. Major implementations of CAR-T cell therapies for tumor treatment are limited by two main factors, the target miss effect, since target antigens could not be highly expressed in tumor cells or be present in normal cells, and the over-activation of immune system, that could induce T-cell death and excessive cytokine production, resulting in nausea, fatigue, anorexia and high fever [25,126]. However, the CAR-T cells therapeutic approach showed promising results for treatment of aggressive B-lymphoma and B-cell precursor acute lymphoblastic leukemia, with two CAR-T cells based viral therapies approved by the European Commission for treatment of hematological neoplasms, tisagenlecleucel (Kymriah, Novartis) and axicabtagene ciloleucel (Yescarta, Gilead) (reviewed in [127]). Despite the enthusiasm of the scientific community, the associated costs limit its widespread implementation (reviewed in [28,128]). Another limitation is the need of large-scale production of viral vectors and associated quality control performed by highly competent technicians [28,128]. To surpass these bottlenecks, non-viral technologies, including Sleeping Beauty and PiggyBac transposon-based vectors [129,130], pDNA transfection [131,132], or different nanoformulations (reviewed in [35]), are being pursued.

#### 2.7.3. Cytokine Genes

The fundamentals of cytokine gene therapy relies on the increase of cytokine levels with anti-tumor properties, including interleukin-2 (IL-2), IL-4, IL-6, IL-12, IL-24, interferon-alpha (IFN-α), IFN-γ, IFN-β or tumor necrosis factors (TNF) TNF-α and TNF-β [25]. The interaction of IL-12 with its receptor results in activation of the JAK-STAT signaling pathway, and activation of IFN-α, with consequent activation of innate and adaptive immune responses [133]. The severe toxic effects experienced by cancer patients after systemic administration of IL-12 lead to the development of in vivo and ex vivo approaches using viral and non-viral vectors to induce expression of the cytokine at the TME (reviewed in [133]). Regardless of the described challenges for gene induced expression mediated by nanoparticles, that end up in modest antitumor effects, the observed severe toxicity related with increased IL-12 concentration in serum triggered the re-focusing towards anticancer therapies that combine the effect of IL-12 with other antitumor strategies, e.g., synergistic effect of IL-12 with other cytokines, such as TNF-α, or GM-CSF, using anti-angiogenic factors, such as VEGF inhibitors, suicide gene therapy or chemotherapy [133,134,135].

### 2.8. Targeting Angiogenesis

The hypoxia experienced in the tumor induced by the uncontrolled growth of tumor cells, induce the secretion of angiogenesis signals, such as vascular endothelial growth factor (VEGF), fibroblast growth factor-2 (FGF-2), angiopoietins or IL-8, to assure oxygen and nutrient supply [136,137]. Two major strategies are being pursued to tackle tumor angiogenesis, 1) down-regulation of pro-angiogenic factors expression, such as VEGF; and 2) up-regulation of expression of anti-angiogenic factors, such as angiostatin, endostatin or TSP-1s (reviewed in [25]). The potential use of angiogenesis targeting for cancer treatment is mainly focused on the administration of engineered antibodies that interfere with angiogenic signals and is limited by the complexity of angiogenic pathway (reviewed in [137]). Indeed, targeting one angiogenic key player could induce other angiogenesis pathways or even induce alternate endothelial-like vascular channels [137].

### 2.9. Targeting Cancer Associated Fibroblasts

Inflammation at the TME renders cancer as a “wound that never heal”, inducing the differentiation of fibroblasts into myofibroblasts, termed as CAFs in the tumor context [6,138,139]. CAFs are a heterogeneous population resultant from different stimulus at the TME including local hypoxia, oxidative stress and growth factors secreted by tumor cells and cells from the immune system (reviewed in [140]). Regarding tumor progression, CAFs stimulate the growth of tumor cells, induce an immunosuppressive TME and stimulate an increased desmoplasia of the ECM [139,140]. Several anti-CAF immunotherapeutic approaches were proposed in the last years for cancer therapy, including elimination or silence of the fibroblast activator protein+ (FAP+) or targeting of the CAF-derived ECM proteins and associated signaling pathways (revised in [140]). FAP is a type 2 dipeptidyl peptidase expressed in CAFs in most solid tumors but also have important roles in the maintenance of normal muscle mass and hematopoiesis [141]. Hence, while FAP targeting CAR-T cells therapy resulted in tumor regression due to enhanced anti-tumor immunity, it also may cause failure of immunosurveillance and alterations in normal tissues, resulting in lethal toxicity anemia and cachexia [84,85,86].

### 2.10. Targeting Tumor Cells Derived Exosomes

Exosomes are nanovesicles synthesized in the endosomal pathway of cells with important roles in inter-cellular communications. They are composed by a lipid membrane and an exosomal lumen composed by proteins and nucleic acids, including mRNA and miRNAs, and their content is dependent of the cell of origin as well as in its physiological condition [142]. Importantly, after internalization by a secondary cell, exosomes induce phenotypical alterations dependent of the exosomal cargo [16,143,144]. Generally, tumor cells secrete higher quantities of exosomes than normal cells, and tumor cells derived exosomes promote tumor progression by inducing malignant transformation in normal cells, tumor escape to immune system, CAF transformation, angiogenesis and metastasis [7]. Hence, efforts are being made to inhibit tumor cells derived exosomes release and uptake [145]. Interestingly, silencing in melanoma cells of Rab27, a protein involved in the transport of the late endosome from the nucleus to the plasmatic membrane, induced miR-494 accumulation, with consequent suppression of malignant phenotype by apoptosis induction [11]. Exosomes can also be used as antigens for tumor vaccination and inhibit cancer progression. An interesting study of Squadrito et al. described a lentivirus-based extracellular vesicle internalizing vector (EVIR) that promoted the selective uptake of extracellular vesicles by DCs that successfully presented the tumor antigens to T cells [146].

## 3. Nanoparticles for Gene Delivery: Fostering Gene Therapy

Cationic lipids, such as Lipofectamine, and biocompatible polymers, are broadly used for intracellular nucleic acid delivery due to their transfection efficacy and ease of production in large-scale. Nevertheless, their low storage stability, lack of targeting capability, and reduced in vivo monitoring, limiting their application in the clinics [19]. Also, the limitations of viral vectors, such as immunogenicity, insertional mutagenesis, poor selectivity and poor efficiency of delivery, lead to the design and development of additional delivery systems. As explained in Section 1, nanoparticles emerged as promising vectors for gene and drug delivery. One of the major advantages of nanomedicines in cancer is that these nanosystems use the tumor tissue physiopathology characterized by a poor lymphatic drainage and a leaky vasculature, with broader fenestrations, facilitating the extravasation of nanoparticles from the surrounding vessels into its interior. This abnormal structure leads to an increased vessels’ permeability and accumulation of nanoparticles in the tumor by passive targeting–EPR effect [19]. Additionally, the ease of functionalization of these nanostructures with different biocompatible molecules, such as PEG and targeting moieties (e.g., antibodies) promotes the active targeting of these moieties to the specific cancer cells with low toxicity [19]. The following section summarizes the most used nanovectors for gene delivery, their advantages and disadvantages and applicability for cancer therapy (resumed in Table 1 and Figure 4).

### 3.1. Inorganic Nanoparticles

Inorganic nanoparticles, due to their low cost, ease of synthesis and good tolerance in biological systems makes them as one of the most used type of nanomaterial employed in nanomedicine, namely as carriers for the cellular delivery of various moieties such as drugs, genes and/or proteins [156].

#### 3.1.1. Metallic Nanoparticles

One of the most used metals in biomedicine is gold due to its benefits in treating inflammation, infection and tuberculosis [157]. Gold nanoparticles (AuNPs) can be easily synthesized using distinct protocols (the most frequent is the reduction of HAuCl_4_), attaining different sizes and shapes like nanorods or nanoshells [158]. Moreover, they can be easily functionalized with different moieties improving biocompatibility and internalization and their optical properties at the nanoscale makes possible to track their intracellular localization [17,158,159]. In the last years, several different applications of AuNPs as carriers in cancer therapy have been described [17,160,161,162]. Indeed, AuNPs functionalized with novel drugs/compounds have been described to increase drug efficacy and tumor reduction [49,160,163,164]. Recently, Coelho et al. developed a drug delivery nanosystem based on pegylated AuNPs loaded with doxorubicin and varlitinib, an anthracycline and a tyrosine kinase inhibitor respectively, for a combined approach against pancreatic cancer cells [165]. AuNPs have been also applied for simultaneous gene and antimicrobial therapy by Peng and collaborators, by conjugating antimicrobial peptides with cationic AuNPs for gene delivery to mesenchymal stem cells [166].

The delivery of TNAs to cells has been a focus of high expectations due to the possibility to treat many human diseases by giving a functional copy of a defective gene or by delivering miRNA, shRNA, ASOs and siRNA to cells [36]. The ideal transfection reagent must protect TNAs from nuclease degradation allowing their release within the nucleus. That is one of the advantages of binding nucleic acids to AuNPs’ surface since, due to steric hindrance the nucleic acid is protected from degradation by nucleases [160]. AuNPs are progressively being used in vitro and in vivo for gene therapy purposes due to their high payload (due to large specific surface area), low toxicity, enhanced uptake, fast endosomal escape, increased half-life; efficient and selective gene silencing [160,167]. For instance, Ryou and collaborators used AuNPs to deliver RNA aptamers, specific to the β-catenin gene, into the nucleus of cancer cells. This strategy, efficiently promoted the inhibition of β-catenin transcriptional activity in the nucleus of lung cancer cells, inducing apoptosis [168]. Moreover, AuNPs have been used as vectors for siRNA delivery, which do not need genome integration for its action, interacting with high specificity to its target and promoting a silencing complex [19,169,170,171]. Additionally, different types of functionalization, like cationic quaternary ammonium or cationic lipid bilayer, allows a more effective siRNA delivery [172]. Some of the obstacles that have limit the application of siRNA conjugated with AuNPs is their aggregation after binding with nucleic acids, reducing their efficacy. Consequently, Elbakry et al., designed a new assembly procedure that consisted in the deposit of siRNA on gold in a layer-by-layer approach [173]. This technique increased the specificity of silencing activity and increased the size uniformity [173]. Furthermore, gold nanorods were used to decrease the expression of some proteins as DARPPP-32, ERK and protein phosphatase in the dopaminergic signaling pathway in the brain, which represent a change in some cancer and drug addiction therapies [174].

Gold nanoconjugates conjugated with oligonucleotides have also demonstrated their effective application in gene therapy [19]. Indeed, Vinhas and collaborators have demonstrated that AuNPs functionalized with an antisense oligonucleotide against *BCR-ABL* mRNA, a fusion mRNA that when translated gives rise to a constitutively active tyrosine kinase that plays a central role on leukemogenesis, induces an effective silencing and increase in K562 cell death [15]. Also, Cordeiro and collaborators demonstrated the applicability and efficiency of Au-nanobeacons for in vivo silencing of a fli-enhanced green fluorescence protein (fli-EGFP) transgenic zebrafish embryos [175].

Abrica-Gonzalez and collaborators analyzed the efficiency of DNA transfection in HEK-293 cells using AuNPs functionalized with chitosan, acylated chitosan and chitosan oligosaccharide. The highest efficiency was obtained with the chitosan oligosaccharide nanoconjugates [176].

Another important type of inorganic nanoparticles used in cancer are iron oxide nanoparticles (IONP) and the superparamagnetic iron oxide nanoparticles (SPION) (reviewed in [177]). In both cases, when an external magnetic field is applied, the particles are attracted to it resulting in the modification of their distribution in the organism [177]. As AuNPs, iron oxide nanoparticles have low toxicity, efficient biodegradability, low cost of production and ease of surface modification [177]. Iron oxide nanoparticles are mainly synthetized by iron coprecipitation in water, obtaining an iron oxide nucleus, and as in the case of AuNPs, they should be covered by amphipathic molecules to improve the biocompatibility [177]. Then, the nanoparticles can be capped with genetic material and allowed to interact with the target cells [177]. There are, however, other methods to synthesize the iron oxide nanoparticles such as reverse micelle mechanism and chemical vapor condensation [177].

There are several applications of iron magnetic nanoparticles in cancer diagnostics and therapy [178,179,180]. Traditionally these nanosystems are used as contrast agents to improve magnetic resonance imaging [181]. Nevertheless, their possible application as carriers in gene therapy is increasing [182,183]. To improve internalization and lysosomal release, functionalization with PEI has been used [184]. However, other authors proposed to transfer DNA coated by nude nanoparticles using an intelligent colloidal nanovector for transfection in equine peripheral blood-derived mesenchymal stem cells with success [185]. Moreover, the efficiency of DNA transfer increases using a magnetic field which delivers the nanoparticles through the cell compartments increasing the DNA delivery efficiency [186]. For this reason, magnetic iron nanoparticles loaded with DNA were employed in mitochondrial therapies with the objective to induce cell death by interacting with the mitochondrial translocation protein [186]. Kim and co-workers evaluated the effect of magnetism and gene silencing strategies by using SPION. These authors demonstrated that the application of a magnetic field that deliver the carrier to an adequate location, increases the efficiency of transfection and promoted the induction of the intrinsic apoptotic pathway [186]. Furthermore, iron nanoparticles loaded with siRNA for silencing gene therapy could be developed due to the little size and the variability of functionalization which provides a net positive surface charge that increases the effect of siRNA. Recently, nanoparticles synthesized with Fe_3_O_4_ were used to target B-cell lymphoma-2 (BCL2) in Ca9-22 oral cancer cells, and the combination with magnetotherapy was able to potentiate the gene silencing effect [184]. Table 2 lists the most recent applications of these metal-based nanoparticles in cancer therapy.

#### 3.1.2. Silica Nanoparticles

Mesoporous silica nanoparticles are normally synthesized from tetraethylorthosilicate (TEOS), as a precursor of silica, and cetyltrimethylammonium bromide (CTAB), as pore generating agent, and reduction conditions using sodium hydroxide and temperature [189]. The major advantages of these nanoparticles are their high surface due to channel formation in their structure and the silanol group in the surface that provides a positive charge for functionalization with nucleic acids and drugs [190]. Bhakta and collaborators demonstrated the application of silica nanoparticles in drug delivery systems using in vitro cells models due to the sedimentation of particles with charge near the cells, which facilitate the integration of genetic material [191]. Indeed, the authors synthetized silica NPs modified in the surface to coat DNA and evaluated the DNA transfection efficiency using in vitro models like COS-7 and 293T cell lines with good rates of transfection [191]. Nevertheless, more recently, Murugadoss and collaborators showed that silica nanoparticles are able to induce toxicity and proinflammatory effects associated with the macrophage activation that reduced its possible application as carriers of genetic material [192].

#### 3.1.3. Carbon-Derived Nanoparticles

Nanoparticles formed by carbon in their structure are an alternative for drug delivery. Carbon nanotubes (CNTs) are molded by one or various sheets of carbon atoms that surround a hollow tubular structure where the genetic material could be deposited [156]. In fact, this structure makes possible to include higher amounts of genetic material because its configuration has a higher surface [156]. The predominant method for synthesis is laser ablation and the size could be adapted using ultrasounds [156]. CNTs could cross across biological membranes and being internalized by the cells by their cape facility, consequently, they are applied in biomedical research as biomolecular carriers [156]. In 2011 Al-Jamal et al., proposed the first application in vivo about the potential use of CNTs to cross neuronal membrane [193]. These authors demonstrated that CNTs with positive charge could transport RNA and reduced in rat brains the synthesis of the protein caspase-3, which is involved in the execution-phase of cell apoptosis [193]. Nevertheless, the application of CNTs is limited by its toxicity [194,195]. In the last years, some alternatives in the functionalization have been developed to reduce the toxicity associated with CNTs, for example, by the introduction of amino group in the nanoparticle structure to decrease their hydrophobic character [194,195].

Nanoparticles based in another carbon configuration are graphene nanoparticles, characterized by a hexagonal monolayer form. These nanoparticles provide a high surface to bind the genetic material and might provide thermal and electric properties [196,197]. To link the nucleotides, the oxidized graphene should be conjugated with PEI or PEG, reducing their toxicity [198]. Their thermal properties could be combined with photothermal therapies in order to facilitate the transfection [199]. Huang el al, showed that PEI-graphene nanoparticles coupled with siRNA were able to tackle Chemokine receptor type 4 (CXCR4) in breast cancer cells (MDA-MB-231) by decreasing not only its mRNA levels but also protein levels [200].

### 3.2. Organic Nanoparticles

There are different types of organic nanoparticles depending of their synthetic procedure and structure. Organic nanoparticles have demonstrated to be able to improve some dose-dependent toxicity associated with other carriers [201].

#### 3.2.1. Lipid Based

Lipid-base systems are the most used in pharmaceutic sciences due to their amphiphilic nature which make possible the interaction with cell membranes and delivery of different type of molecules/compounds like vitamins, A, E and D and drugs [201,202]. The efficiency of incorporation of the different molecules and compounds will depend on their composition, that will affect also the delivery efficiency [203].

Liposomes are lipid vesicles that can incorporate drugs or genetic material inside (hydrophilic compartment) or in the hydrophobic side (lipidic membrane) depending on the hydrophilic or hydrophobic characteristics of the molecules, respectively. Different methods can be used to synthesize liposomal formulations and for their characterization and numerous patented formulations have been described (for a more comprehensive review on this see [204]). The delivery of cargo in non-targeted liposomes is mediated by the EPR effect [205]. Nevertheless, in the last years, targeted liposomal formulations have been described and particularly folate-conjugated liposomes that involves the attachment of folic acid with phospholipids, cholesterol or peptides before liposomal formulation synthesis has been developed (for a review see [206]). The most common application of liposomes in gene delivery is to carry DNA into the cells [22]. In some cases, to increase the transfection efficacy, DNA-liposome therapy is combined with other mechanic procedures like ultrasounds [22]. Chen and collaborators designed a liposome-based nanoconjugate (p53/C-rNC/L-FA) for site-specific intracellular delivery of an apoptotic protein cytochrome c and a plasmid DNA encoding tumor-suppressing p53 protein [207]. The authors demonstrated that p53/C-rNC/L-FA liposome induce apoptosis in tumor cell lines and inhibited tumor growth in a breast cancer mouse model [207]. Zuo and collaborators also used liposomes to deliver p53 but in this case for ovarian cancer targeted therapy [208]. Indeed, they synthesize cationic polymeric liposomes composed of an EGF derivative (EGF-GHDC), cholesterol, and DOPE, for the systemic delivery of the p53 gene to ovarian cancer cells with high efficiency [208].

Liposomes are also one of the most widely used carriers for CRISPR/Cas9 delivery [209]. Recently, Hosseini and collaborators described the synthesis, characterization and in vitro effect of a cholesterol-rich lipid-mediated nanoparticles for the transfection of Cas9/sgRNA plasmids [210]. Their liposomal formulation was capable to boost transfection in HEK293 cell line stably expressing GFP and efficiently knockout GFP expression [210]. For additional details on non-viral delivery systems, including liposomal formulations, for CRISPR/Cas9-based genome editing please see [209,211,212]).

Palazzolo and collaborators described the in vivo effect of a DNA origami that was designed to fit inside a stealth liposome to deliver doxorubicin [213]. They demonstrated that this advanced drug delivery system improved the antitumoral efficacy of doxorubicin in tumor-bearing mice and decreased the DNA origami toxicity [213].

Recently, Shao and collaborators used a liposomal nanoformulation to efficiently target TMPRSS2/ERG fusion mRNA and to enhance docetaxel treatment in prostate cancer [214]. Al-Attar and collaborators evaluated a combination of drug delivery devices composed of holo-transferrin conjugated liposomes for siRNA (targeting BCR-ABL) delivery, and electrospun polycaprolactone-gelatin microfibers for resveratrol release [215]. For additional applications on liposomes for gene therapy see [50,209,216,217,218,219].

As an alternative to liposomes, solid-lipid nanoparticles (SLNs) are also able to protect the active cargo from degradation and make possible the modulation of delivery. SLN is a solid lipid capped by a layer of surfactants in aqueous dispersion. There are numerous synthesis procedures, the most common used is high pressure homogenization [153,220]. To promote the functionalization with the genetic material several types of surfactants might be used. Lipid carrier nanoparticles have applied in the treatment of ocular diseases and infections [221,222,223]. Lipids nanoparticles charged with drugs or a condensed DNA core surrounded by lipid bilayers have been applied in retinitis pigmentosa or age-related macular degeneration [221]. The combination of DNA delivery using SLN and drugs was proposed as a promising strategy for cancer therapy [153]. Penumarthi and collaborators developed a SLN conjugated with DNA and showed the biocompatibility and the high transfection rate in dendritic cells [224]. Table 3 shows the most recent applications of these lipid-based nanoparticles in cancer therapy.

#### 3.2.2. Polymeric Nanoparticles

Polymeric nanoparticles are obtained from preformed polymeric materials or from monomeric structures [225]. Due to their biodegradability, compatibility and controlled release, the natural biopolymers have increased the attention in developing new biomedical tools [225,226,227]. Cationic polymers have positive charges in their structure and could interact between anionic genetic material and could bind to DNA forming a complex knows as polyplexes, which have small size and less degradation than other polymers [228]. Other polymers used in therapy are biopolymers which have been synthetized from different natural sources like cellulose, starch or algae [229,230,231]. These polymers have excellent biocompatibility, biodegradation, low toxicity and good mechanic properties, making them a promising tool in gene therapy systems [226]. However, polymer-based nanoparticles have less efficacy than other particles for genetic material delivery [229]. In an effort to improve the capacity of polymeric nanoparticles to overcome the intracellular barriers and consequently increase the transfection efficacy, Santo et al. constructed poly(2-aminoethyl methacrylate) (PAMA)-based polyplexes covered with PEG and evaluated the efficiency of in COS-7 and HepG2 cell lines. The results showed that the nanosystem covered with PEG increased the transfection activity and decreased the cytotoxicity [232]. Zhupanyn and co-workers evaluated the application of PEIs nanoparticles conjugated with extracellular vehicle (ECV) to deliver siRNA in different cell lines. They demonstrated high knockdown efficacy in mrR-155 and miR-1246 genes of the PEI/siRNA modified with ECV. These results could be associated with the PEIs structure which makes possible the formation of non-covalent complexes with RNA molecules [233]. Table 4 summarizes the most recent application of polymeric nanoparticles in cancer therapy.

### 3.3. Biological Nanoparticles

The need to pursue novel and effective therapeutic strategies for cancer treatment has led to the exploration of biological vesicles for gene delivery [238]. The intercellular communication role of exosomes makes them suitable nanovectors for gene therapy [239]. The engineering of allogenic exosomes with miRNAs, mRNAs, proteins for targeted delivery, or drugs, allow the delivery of the therapy to the cells of interest with a reduced immune response, overcoming major shortcomings faced by gene delivery [155]. The manipulation of exosome content may be accomplished by two approaches, engineering of parent cells to secrete modified exosomes, or directly modify the exosome content after secretion (reviewed in [155]). The direct modification of exosomes could be achieved by e.g., incubation, electroporation or lipofection [155]. Some limitations of these methodologies include restricted of the cargo type and size when using incubation, poor efficiency of DNA transfer when using electroporation, and low transfection efficiency when using lipofection [155]. In a study performed by Kim et al., macrophage derived exosomes were loaded with paclitaxel by incubation, electroporation and sonication [240]. The sonication methodology produced exosomes with higher paclitaxel loading efficiency, and consequent higher drug release in vivo [240]. The genetic engineering of parent cells may be pursued by using the methodologies here described. However, careful must be taken regarding the choice of the donor cell, as exosomes content reflect the cell of origin and potential negative effects can occur if tumor cells derived exosomes are used [155]. Table 4 summarizes current studies that used exosomes for gene delivery. After profiling the exosome miRNAs secreted by different populations of triple negative breast cancer (TNBC) cells, O’Brien et al. observed a decreased abundance of miR-134, involved in regulation of STAT5B that regulates the heat shock protein HSP90 [241]. Genetic engineered miR-134 overexpressing Hs578Ts(i) cells secreted exosomes with higher amounts of miR-134, that reduced the expression of STAT5B and Hsp90 in Hs578Ts(i) recipient cells, with consequent decrease of cellular migration and increased sensitivity to anti-Hsp90 drugs [241]. In another study, Lou et al. transfected adipose tissue-derived mesenchymal stem cells (AMSC) with miR-122 expression plasmid and harvested secreted exosomes [242]. Treatment of hepatocellular carcinoma cells with miR-122 enriched exosomes resulted in an increased cell chemosensitivity to sorafenib in vivo [242]. In a recent study, Yuan et al. treated the TNBC cell line MDA-MB-231 with exosomes secreted by human umbilical cord mesenchymal stem cells overexpressing miR-148b-3p (HUCMSC-miR-148b-3p) [243]. An inhibitory effect on MDA-MB-231 proliferation was observed after treatment with HUCMSC-miR-148b-3p derived exosomes, highlighting the potential use of miR-148b-3p containing exosomes for breast cancer treatment [243]. Gong et al. took advantage of exosomes as endogenous nanocarriers and analyzed the potential of co-delivery of doxorubicin and hydrophobically modified miR-159 for treatment of triple-negative breast cancer [244]. Exosomes from macrophages resultant from differentiated THP1 monocytes cell culture were loaded by incubation with doxorubicin and Cho-miR159 [244]. Treatment of TNBC cells with the exosomes resulted in an improved anti-cancer effect [244]. Table 5 highlights some strategies using exosomes for gene therapy.

## 4. Current Trends in Nanovectorization of Gene Therapy: A Focus on Cancer

As the knowledge of cancer progresses, it is becoming clear that nanovectorization of genes for cancer therapy is pivotal for effective gene delivery with low toxicity. Following chapter will address the efforts that will allow the transfer of the nanomedicines from bench to bedside.

### 4.1. Translating to the Clinics

The advent of precision medicine assisted by gene targeted delivery via nanoparticles allowed to tackle cancer with higher efficacy. However, the application of nanotechnology for cancer gene therapy is still in its infancy and only a few phase I and II clinical trials were projected, mainly based on organic nanoparticles (Table 6). It is not surprising that most of the clinical trials are focused on organic nanoparticles such as liposomes or solid-lipid (Table 6), since drug-liposomes formulations are currently approved by the FDA for cancer therapy, including Doxil (US) or Caelyx (EU), pegylated liposomes with doxorubicin, or Myocet, a non-pegylated liposome with doxorubicin [249]. As summarized in Table 6, gene therapy formulations based on liposomes focus on the targeting of proto-oncogene *cRAF* with or without adjuvant chemotherapy (NCT00024661 and NCT00024648), delivery of tumor suppressor genes (*FUS1,* NCT00059605 and *TP53,* NCT02354547), or for the delivery of mRNA vaccines for treatment of ovarian cancer (NCT04163094) or melanoma (NCT02410733). Phase 1 clinical trial NCT00059605 reported a safe intravenous administration of lipoprotein DOTAP:chol-FUS1 to lung cancer patients with a MTD of 0.06 mg/Kg, that resulted in an effective alteration of TUSC2-regulated pathways and consequent anti-tumor effects, with five patients in a 31 patients cohort achieving stable disease after at least 2.6 months [250]. In another Phase I clinical trial (NCT00938574) Schultheis et al. described a liposome-based RNA interference therapy–Atu027–consisting in liposomes containing siRNA that silence expression of protein kinase N3 in vascular endothelium [251]. The formulation was administered in 34 patients with advanced solid tumors with a 0.366 mg/kg tolerability with only low-grade toxicities observed, resulting in disease stabilization in 41% of the patients after at least 8 weeks [251]. The phase I/phase II clinical trial NCT02191878 was designed to examine the safety, pharmacokinetics and antitumor activity of the TKM-080301 formulation based on LNPs carrying siRNA directed against human PLK1 designed for intravenous delivery [252]. The study involved 43 patients with hepatocellular carcinoma showing tolerability to the drug with a MTD of 0.6 mg/kg [252]. Despite the observed stable disease in 23.1% of the subjects after 7.5 months, authors concluded that further application of the drug only results in modest antitumor efficacy and the efficacy of TKM-080301 as single agent was not enough to be further explored [252].

In the saga to find an effective personalized gene therapy, mRNA vaccines are being tested in clinical trials using autologous DCs (NCT02882659), DCs transfected in vitro with tumor mRNA (NCT01278914) or mRNA tumor antigen pulsed DCs (NCT02808416). However, few phases I clinical trials were projected to analyze the efficacy of mRNA vaccines delivery using nanoparticles (Table 6). The OLIVIA clinical trial (NCT04163094) is a phase-I study in ovarian cancer patients that consist in the treatment with neoadjuvant chemotherapy with carboplatin and paclitaxel, combined with W_ova1 mRNA, a liposome-based vaccination. 

The KEYNOTE-603 phase I clinical trial (NCT03313778), intends to analyze the safety, tolerability and immunogenicity of mRNA-4157 alone or combined with pembrolizumab for treatment of unresectable solid tumors. The mRNA-4157 is a personalized mRNA-based vaccine, designed after identification of twenty patient specific epitopes that are transcribed and loaded on a single mRNA molecule in a solid-lipid vesicle [254]. The mRNA translation by APCs allowed the induction of T-cells that target the patient’s tumor-specific epitopes [254]. The favorable clinical responses, consisting in mRNA-4157 tolerability at all dose levels tested, suggested the possibility of further clinical trials [254].

Despite only one study using metallic nanoparticles is under clinical trials (NCT03020017, Table 6), the potential of inorganic nanoparticles for tumor vaccination is supported by studies from the University of Minnesota (USA) that conducted phase II clinical trials using allogenic large multivalent immunogen (LMI) vaccines in silica particles combined with IL-2 for stage IV melanoma and breast cancer treatment (NCT00726739 and NCT00784524) [255]. Nevertheless, the clinical translation of inorganic nanoparticles is hampered by difficulties in consistent and reproducible scale-up and lack in controlling their long-term biological fate [256]. These bottlenecks are, in fact, a few of the major concerns that limit the application of nanomedicine in preclinical and clinical studies, that include: (1) batch-to-batch reproducibility, (2) scalability of nanoparticles production, (3) incomplete information on the biological effect of nanoparticles in vivo, (4) limited increase of therapeutic indexes, (5) reluctance of the pharmaceutical industry to invest in nanomedicine, (6) lack of regulatory issues and safety from authorities and regulatory agencies [257,258]. Furthermore, a recent review of Ioannidis et al. discusses the necessity of more thoughtful design of preclinical studies to potentiate the hypothesis for translation to the clinics [259]. According to the authors, the applied research practices result in biased and low reproducible preclinical studies, with poor design of the studies, poor characterization of the nanomaterials, incomplete studies of their biological effect, models heterogeneity and poor use of controls and statistics [259].

### 4.2. Combined Therapies

The tumor complexity, especially in advanced cancer stages, render combination therapies as obligatory for anticancer therapies. The application of other therapies, such as chemotherapy or radiotherapy, with gene therapies is hampered by the limited efficacy of the later [260]. In a phase I clinical trial, Matsumoto et al. studied the efficacy of cationic liposomes containing the human interferon beta (HuIFNβ) in patients with advanced melanoma [261]. Despite no adverse effects of the therapy were observed, the efficacy of the gene therapy was insufficient [261]. Three different clinical trials studied the effect of TKM-080301 (Table 6), a SLN formulation containing siRNA against Polo-like kinase 1 (*PLK1)* gene, a serine/threonine kinase involved in cell cycle progression and mitosis. The studies evaluated the nanoformulation for treatment of liver cancer (NCT01437007), adrenocortical cancer (NCT01262235) and hepatocellular carcinoma (NCT02191878). The results from clinical trials evaluating the efficacy of the therapeutics in the two last studies revealed good tolerability, but while it was observed an anticancer efficiency in adrenocortical cancer patients, a limited efficacy was observed in hepatocellular carcinoma patients [253,262].

The design of clinical studies focus on the effect of the gene therapy combined with chemotherapeutic drugs that act on TME (Table 6), such as the immune checkpoint inhibitors pembrolizumab, and durvalumab, that consist of antibodies directed against programmed death 1 receptor (PD-1), or that block the interaction between PD-1 and its ligand (PD-L1), respectively. The phase I clinical trial NCT03323398, aims to induce the immune system by analyzing the efficacy of durvalumab with mRNA 2416 formulation, a SLN encapsulating the tumor necrosis factor ligand superfamily member 4, OX40 ligand, that ultimately induce the proliferation of T lymphocytes. The study NCT03948763 combine the immune checkpoint inhibitor pembrolizumab, with a lipidic nanoformulation containing mRNA of the most frequent mutations of the oncogene *KRAS*, that after internalization by APCs induce T-lymphocytes that specifically target and destroy tumor cells harboring this KRAS mutations.

Other anti-tumor strategies use gene-based therapy to simultaneously inhibit the maturation of TME and inhibit tumor cell proliferation. The phase I clinical trial NCT01158079, consisted in a solid-lipid nanoformulation, named ALN-VSP02, containing siRNAs that target VEGF-A gene, to inhibit neo-angiogenesis, and the kinesin spindle protein (*KSP*) gene, leading to cell cycle arrest [263]. After treatment of thirty-one patients with advanced solid tumors, it was observed tolerability to the therapeutics and anti-tumor activity, consistent with an anti-VEGF effect [263].

## 5. Conclusions

Nanomedicine has been providing conceptual solutions to the limitations faced by viral-based gene delivery. The effort to design biocompatible and non-immunogenic nanoparticles has been gradually showing some conceptual success, with some of these being further promoted to clinical use. A plethora of nanomedicines have continuously been proposed for efficient vectorization of TNAs against deregulated cancer mechanisms. However, while in vitro proof-of-concept studies are a useful prerequisite, early progression to in vivo testing of novel nanomedicine approaches is to be encouraged so as to improve their potential clinical utility. Due to the low immunogenicity and easy penetration of the cell membrane, lipid-based nanoparticles and exosomes are, perhaps, two of the most promising vectorization strategies for gene delivery, reflected by the number of proposed clinical trials using these nanovectors for cancer. Indeed, the clinical studies under way (Table 6) highlight the potentiality of nanomedicine to improve the miRNA- and siRNA-based therapeutics. However, the effective application of nanomedicine in the clinics is hampered by a poor study design, being required more thoughtful studies with good characterization of nanoparticles and their biological effect. Nevertheless, the tumor heterogeneity and TME complexity still poses limitations for an effective treatment focusing on gene therapy, being required the formulation of novel strategies using combined anti-tumor therapies.

## Figures and Tables

**Figure 1 pharmaceutics-12-00233-f001:**
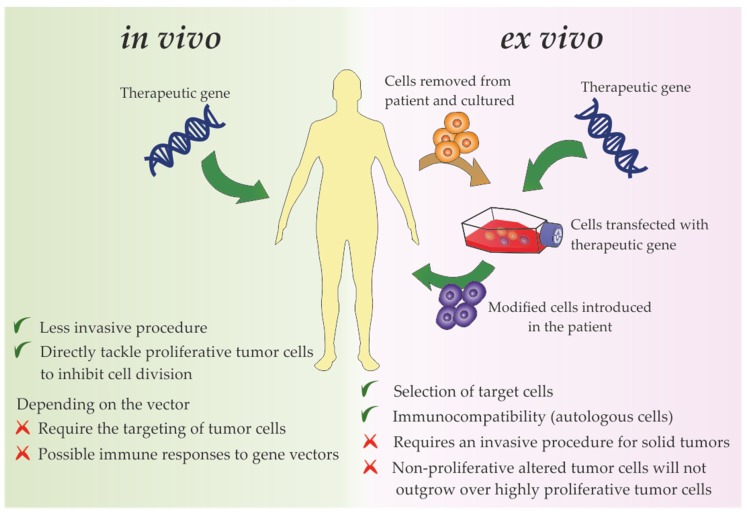
Delivery strategies used for gene therapy directly targeting tumor cells or tumor microenvironment components, their major advantages (preceded by a green checkmark) and disadvantages (preceded by a red cross).

**Figure 2 pharmaceutics-12-00233-f002:**
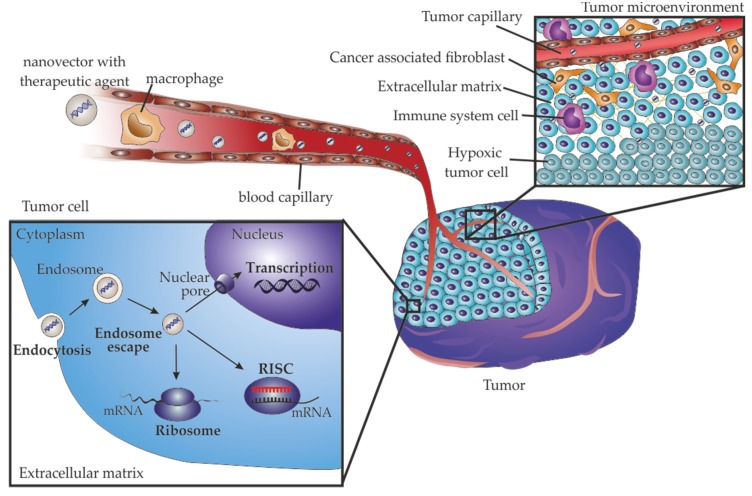
Barriers that nanoparticles must overcome for effective cancer gene delivery. In a systemic administration, nanoparticles should travel through the blood circulatory system, avoiding the immune system. The accumulation at the tumor occurs through passive targeting by the enhanced permeability and retention effect. Nanoparticles also have to penetrate into the most inaccessible areas of the tumor to reach the hypoxic tumor region with low oxygenation and dense extracellular matrix. After reaching tumor cells, nanoparticles should be internalized, which is mainly accomplished via endocytosis, and then escape from the endosome to efficiently deliver the cargo into the cytoplasm, when targeting RNA, or travel to the nucleus, when targeting DNA.

**Figure 3 pharmaceutics-12-00233-f003:**
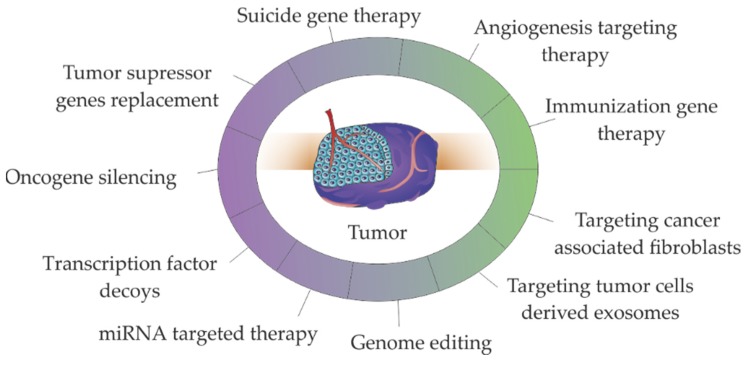
Major strategies used in non-viral gene therapies for cancer treatment. Therapies targeting the tumor microenvironment (in green), including angiogenesis targeting therapy, immunization gene therapy, targeting cancer associated fibroblasts and targeting tumor cells derived exosomes, also use the described molecular strategies (in purple), such as genes replacement, gene silencing, transcription factor decoys, miRNA targeted therapy and genome editing.

**Figure 4 pharmaceutics-12-00233-f004:**
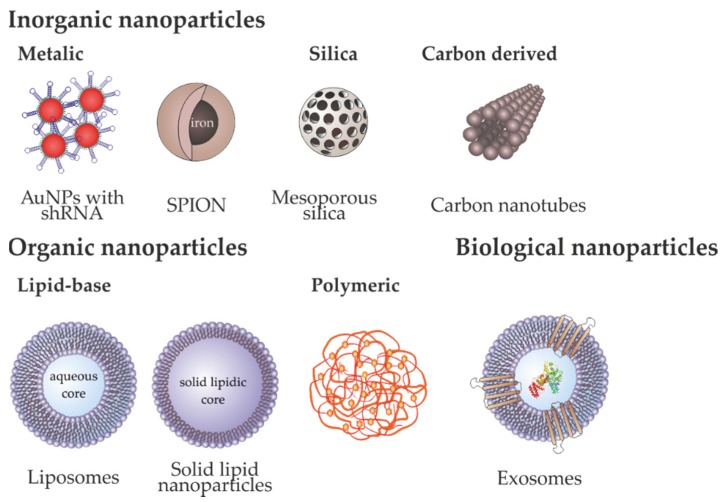
Nanoparticles used for gene delivery. Examples of metallic nanoparticles are gold nanoparticles (AuNPs) that can be functionalized with several molecules, e.g., short hairpin RNA (shRNA) for gene silencing. Other examples of inorganic nanoparticles are superparamagnetic iron oxide nanoparticles (SPION) containing an iron core coated with biocompatible polymers, mesoporous silica nanoparticles or carbon nanotubes. Examples of organic nanoparticles are polymeric nanoparticles, and liposomes and solid lipid nanoparticles (SLNs), which are lipid-base nanoparticles that differ mainly in the aqueous and lipidic core and the number of lipid layers. Exosomes are nanovesicles secreted by eukaryotic cells composed by a bi-lipidic membrane containing membrane proteins, that surround an aqueous lumen containing proteins and nucleic acids.

**Table 1 pharmaceutics-12-00233-t001:** Advantages and disadvantages of nanoparticles used for gene delivery.

Composition	Advantages	Disadvantages	Reference
**Inorganic Nanoparticles**			
Metallic (Gold, iron)	Multiple forms (spherical, nanorods, triangles) Biocompatibility Tunable size Straightforward functionalization	More information about uptake, biocompatibility and low cytotoxicity are required for clinical translation	[19,147]
Silica	The structure could add high amounts of drugs and genes Tunable pore size	More information about cytotoxicity, biodistribution and biocompatibility are required for clinical translation	[148]
Carbon-derived	Large surface area High loading capacity Vast numbers of possibilities for surface modification	Few in vivo studies developed	[149,150,151]
**Organic nanoparticles**			
Lipid-base (Liposomes)	Low toxicity Biodegradable Can transport hydrophobic and hydrophilic molecules	Moderate loading capacity Could crystallize after prolonged storage conditions	[152,153]
Polymeric	Biodegradable properties Good tissue penetration Ease manipulation	Non-degradable polymers tend to accumulate in tissues Promote allergic reactions In vivo metabolism and elimination routes are not elucidated	[154]
**Biological nanoparticles ***			
Exosomes	Reduced immune response Protection of circulating genetic material Possibility of cell targeting	Limited transfection efficiency	[155]

* Viruses were not included since they are out of the scope of the review.

**Table 2 pharmaceutics-12-00233-t002:** Application of metallic nanoparticles in cancer therapy.

Type of Nanoparticle	Description	Implication in Tumor Eradication		Reference
		In Vitro	In Vivo	
Gold	AuNP ^1^ with NLS ^2^-peptide-target, designed to inhibit miR-211 and NCL ^3^ function in AML ^4^ cells	Effective silencing of the NCL/miR-211/NFkB/DNMT1 pathway, with synergistic blockage of AML cell proliferation and colony formation	Extension of AML mice survival, lower white blood cells, metastases to lungs and blasts in bone marrow and reversed splenomegaly	[187]
Gold	AuNPs wrapped in dsDNA ^5^ loops from pEGFP ^6^ to retinal pigment epithelium cells	DNA-AuNPs were faster internalized/uptaken than liposome complexes		[188]
Gold	PEGylated AuNP functionalized with antisense oligonucleotide against BCR-ABL mRNA	Decreased proliferation and cell viability in chronic myeloid leukemia cell lines		[18]
Gold	AuNP functionalized with chitosan oligosaccharide and pIRES_2_-EFGP or pSV-β-Gal plasmids	Green/one-pot synthesis of chitosan-AuNPs without toxic chemical reagents, showed increased transfection of pDNA into HEK-293 cell line.		[176]
Iron	Oxide iron NPs to deliver siRNA targeting BCL-2 in oral cancer cells	Reduced cell viability and relative cell migration in Ca9-22 cell line.		[184]
Iron	SPIONs ^7^ with chitosan-graft-PEI and PK11195 as gene carriers targeting the mitochondria	Reduced cell viability of tumor cell lines A549, KB and Hella, via alteration of mitochondrial metabolism, and under an external magnetic field, increased the apoptotic pathway.		[186]

^1^ AuNP: gold nanoparticle (NP), ^2^ NLS: nuclear localization signal, ^3^ NCL: nucleolin, ^4^ AML: acute myeloid leukemia. ^5^ dsDNA: double stranded DNA. ^6^ pEGFP: enhanced green fluorescent protein plasmid. ^7^ SPIONs: superparamagnetic iron oxide nanoparticles.

**Table 3 pharmaceutics-12-00233-t003:** Application of liposomes and solid-liquid nanoparticles in cancer therapy.

Type of Particle	Description	Implication in Tumor Eradication		Reference
		In Vitro	In Vivo	
Liposome	AS1411 (DNA aptamer targeting nucleolin) and 5-Fluorouracil functionalized liposomes	Functionalized liposomes increased apoptosis of basal cell carcinoma TE 354.T cell line, compared to non-functionalized, with low toxicity in human dermal fibroblasts.		[222]
Liposome	p53/C-rNC ^1^/L-FA ^2^ liposome composed by a fusogenic liposomal shell that promote fusion of lipoprotein with cell membrane under acidic pH, and after degradation of polymeric C-rNC shell by GSH ^3^ in the cytoplasm, release pDNA encoding p53 and CytoC	p53/C-rNC/L-FA liposome induced higher cell apoptosis than other formulations in breast cancer cell line MCF-7	Nanoparticle system decreased tumor weight and volume in mice models after 30 d of treatment with no apparent toxic side-effects. Nanoformulation half-life was 8.8 h and mean residence time was 12.1 h.	[207]
Liposome	P53 gene loaded EGF-targeted CPL ^4^	Significantly enhanced liposome half-life and efficient p53 delivery to ovarian cancer cell line SKOV3.	CPLs containing EGF showed higher tumor-targeting capacity in tumor-bearing nude mice	[208]
Liposome	Design of CRISPR/Cas9 cationic liposomes	PEGylated cholesterol domain lipoplexes optimized for efficient delivery of CRISPR/Cas9 increased the gene-editing efficiency by 39%.		[210]
Liposome	Liposome encapsulating eIF3i ^5^ shRNA ^6^	The liposomes induced a dose-dependent inhibition of cell proliferation and migration in murine melanoma B16F10 cell line	Nanoformulation inhibited melanoma lung B16F10 derived metastasis in male mice	[10]
Liposome	PEG ^7^-modified liposomes containing TMPRSS2/ERG fusion siRNA ^8^ in combination with docetaxel		Liposomes and docetaxel chemotherapy resulted in decrease of prostate tumor growth in mice models after 3 weeks of treatment (92% with respect to control)	[214]
Liposome	Liposomes containing BCR-ABL siRNA combined with Resveratrol loaded in electrospun fibers for controlled release	Combination of liposomes with resveratrol decreased CML ^9^ K562 cell viability in a dose dependent manner		[215]
Liposomes	Design of cationic liposomes to evaluate transfection efficiency of Cas9/sgRNA	Lipid based nanoparticles with cholesterol increased the stability and transfection efficacy for Cas9/sgRNA delivery, resulting in 39% gene-editing efficiency in HEK293 cell line.		[210]
Solid lipid nanoparticles	Application of SLNs ^10^ as non-viral DNA vaccine delivery system	DNA-SLN complexes showed no cytotoxicity and high transfection efficiency to dendritic cells		[224]

^1^ C-rNC: Cyto-C encapsulated redox-responsive nanocapsule. ^2^ L-FA: folic acid. ^3^ GSH: Glutathione. ^4^ CPL: cationic polymeric liposomes. ^5^ eIF3i: eukaryotic translation initiation factor. ^6^ shRNA: short hairpin RNA; ^7^ PEG: polyethylenoglycol; ^8^ siRNA: small interfering RNA; ^9^ CML: chronic myeloid leukemia. ^10^ SLN: solid lipid nanoparticles.

**Table 4 pharmaceutics-12-00233-t004:** Polymeric nanoparticles used in cancer therapy.

Gene Delivery	Description	Implication in Tumor Eradication		Reference
		In Vitro	In Vivo	
siRNA ^1^	Hyaluronic-acid-modified chitosan nanoparticles labeled with Cyanine 3 (Cy3) to deliver *BCL2* siRNA to A549 cell line	Nanoformulations induced inhibition of cell proliferation via *BCL2* downregulation		[234]
siRNA	Analysis of alkylation effect in alkylated cationic curdlan nanoparticles as STAT3 siRNA carriers	Alkylation significantly reduces the cytotoxicity of aminated curdlan in human HepG2 and murine B16 cell lines		[235]
siRNA	Polymeric nanoparticle based on curdlan loaded with FITC or Plk1 ^2^ siRNA in a pH dependent manner	Curimi polymers enhanced endosomal escape and efficiently delivered siRNAs to Hela and HepG2 cells resulting in cell growth inhibition and DNA damage		[236]
pDNA ^3^	Analysis of effect of polymeric particles with pH dependent switching surface charge, carrying FITC-pDNA ^3^	Nanoformulations efficiently transfected Hela cells in a pH dependent manner		[237]
siRNA	Polyethylenimine polymeric particles complexed with extracellular vesicles (EVs) combined for siRNA and anti-miRNA ^4^ delivery	Ddose-dependent inhibition of miRNAs. Differences in the polymer/siRNA efficacies between ECVs from different cell lines, regardless of the target cell line	Xenographts size decreased after 12 days of treatment in mice model after intravenous injection	[233]

^1^ siRNA: small interfering RNA; ^2^ Plk1: Plo-like kinase-1; ^3^ FITC-pDNA: plasmid DNA containing fluorescein-5-isothiocyanate gene. ^4^ miRNA: microRNA.

**Table 5 pharmaceutics-12-00233-t005:** Gene therapy strategies using exosomes as gene delivery vehicles.

Gene Delivery	Description	Implication in Tumor Eradication		Reference
		In Vitro	In Vivo	
miRNA ^1^	Treatment of Hs578Ts cells with exosomes derived from miR-134 overexpressing Hs578Ts cells	Potential of miR-134 as biomarker and therapeutic target for TNBC ^2^ treatment		[241]
miRNA	Treatment of HCCs ^3^ with miR-122 enriched exosomes secreted by AMSC ^4^	Successful packaging of miR-122 in secreted exosomes increasing sensitivity of HCCs to chemotherapeutic drugs	Intra-tumor injection of AMSC miR-122 exosomes increased antitumor efficacy to sorafenib	[242]
miRNA	Treatment of TNBC ^4^ cells with HUCMSC-miR-148b-3p ^5^ derived exosomes	HUCMSC-miR-148b-3p derived exosomes inhibited cell proliferation, invasion and migration, and induced cell apoptosis in MDA-MB-231 cell line	Exosomal miR-148b-3p inhibited formation of tumors and metastasis in nude mice	[243]
miRNA	Treatment of TNBC cells with macrophages derived exosomes loaded with doxorubicin and hydrophobically modified miR-159	Effective uptake of the nanoformulation in MDA-MB-231 cell line, resulting in increased apoptosis and decreased cell migration	Synergistic tumor decreases in MDA-MB-231 xenografted-nude mice, reaching 92.8% overall inhibitory rate of tumor volume	[244]
miRNA	Treatment of NSCLC ^6^ with HEK293T cells derived exosomes transfected with miR-497	Nanoformulations efficiently suppressed tumor growth in a microfluidic 3D lung cancer model		[245]
Anti-miRNA	Targeting of breast cancer stem cells with MSCs ^7^ derived exosomes loaded with LNA ^8^-antimiR-142-3p by electroporation	LNA-antimiR-142-3p internalization by MCF-7 derived MSCs resulted in decreased colony formation capability	Decreased ability of tumor-initiating capability of MCF-7 derived MSCs after internalization of the nanoformulation	[246]
siRNA ^9^	Folate displaying HEK293T cells derived exosomes carrying survivin siRNA	Nanoformulations displayed cytosolic delivery without endosome entrapping		[247]
siRNA	Cationic bovine serum albumin conjugated with siS100A4 and TNBC-derived exosome membrane coated nanoparticles	Nanoformulations efficiently inhibited growth of breast cancer cells	Nanoparticles showed good biocompatibility and higher affinity to lungs than similar nanoparticles coated with liposomes	[248]

^1^ miRNA: microRNA; ^2^ TNBC: triple-negative breast cancer; ^3^ HCCs: hepatocellular carcinoma cells; ^4^ AMSC: adipose tissue-derived mesenchymal stem cells; ^5^ HUCMSC: miR-148b-3p-human umbilical cord mesenchymal stem cells overexpressing miR-148b-3p; ^6^ NSCLC: non-small cell lung cancer; ^7^ MSCs: mesenchymal stem cells; ^8^ LNA: locked nucleic acid; ^9^ siRNA: small interfering RNA.

**Table 6 pharmaceutics-12-00233-t006:** Nanoparticles for gene therapy in interventional clinical trials. Data acquired from the U.S. National Library of Medicine [253].

Nanoparticle Type	Nanoparticle Description + Chemotherapeutic Agent	Cancer Type	Clinical Trial Reference (Phase); Status; Reference
**Inorganic Nanoparticles**			
AuNPs ^1^	NU-0129 – nucleic acids targeting *BCL2L12 ^2^* in spherical AuNPs	Glioblastoma/Gliosarcoma	NCT03020017 (1); Active
**Organic nanoparticles.**			
Liposomes	DOTAP: Chol-FUS1 – DOTAP ^3^:cholesterol liposome with *FUS1 ^4^* gene	NSCLC ^5^	NCT00059605 (1); Completed; [250]
Liposomes	SGT-53 – liposomal nanocomplex for targeted delivery of wild type *TP53* gene + Topotecan + cyclophosphamide	Solid tumors	NCT02354547 (1); Recruiting
Liposomes	SGT-53 – liposomal nanocomplex for targeted delivery of wild type *TP53* gene + temozolomide	Glioblastoma	NCT02340156 (2); Terminated
Liposomes	DOTMA ^6^:cholesterol liposomes with *IL-2* ^7^ gene	Recurrent head and neck cancer	NCT00006033 (2); Completed
Liposomes	LErafAON – c-*RAF ^8^* antisense oligonucleotide encapsulated in liposomes	Advanced solid tumors	NCT00024661 (1); Completed
Liposomes	LErafAON – c-*RAF* antisense oligonucleotide encapsulated in liposomes	Advanced solid tumors	NCT00024648 (1); Completed
Liposomes	*EPHA2 ^9^*-targeting siRNA ^10^ encapsulated in DOPC ^11^ liposome	Advanced solid tumors	NCT01591356 (1); Recruiting
Liposomes	DOTAP:Chol-TUSC2 - – DOTAP:cholesterol liposome with *TUSC2 ^12^* gene + erlotinib	NSCLC	NCT01455389 (1/2); Active
Liposome	W_ova1 vaccine – liposome formulated mRNA ^13^ vaccine	Ovarian cancer	NCT04163094 (1); Recruiting
Liposome	Atu027 – liposome with siRNA against PKN3 ^14^	Advanced solid tumors	NCT00938574 (1); Completed; [251]
Liposome	Atu027 – liposome with siRNA against PKN3 + gemcitabine	Pancreatic ductal carcinoma	NCT01808638 (1/2); Completed
Liposome	Lipo-MERIT – liposomes with tumor-antigen encoding RNAs	Melanoma	NCT02410733 (1); Recruiting
Liposome	TNBC-MERIT – liposomes with tumor-antigen encoding RNAs	TNBC	NCT02316457 (1); Active
Liposome	Pbi-shRNA STMN1 lipoplex – shRNA ^15^ against *STMN1 ^16^* in liposome	Solid tumors	NCT01505153 (1); Completed
Liposome	MTL-CEBPA – saRNA ^17^ targeting *CEBPA ^18^* in liposome	Liver cancer	NCT02716012 (1); Recruiting
Liposome	MTL-CEBPA – saRNA targeting *CEBPA* in liposome + pembrolizumab	Solid tumors	NCT04105335 (1); Not yet recruiting
Liposome	BP1001 – antisense oligonucleotide of *GRB2 ^19^* in liposome + low-dose ara-C	Philadelphia chromosome positive Leukemia	NCT01159028 (1); Active
Solid lipid	mRNA-2416 – lipid nanoparticle encapsulated mRNA encoding human *OX40L ^20^* + durvalumab	Solid tumors/ lymphoma	NCT03323398 (1/2); Recruiting
Solid lipid	mRNA-2752 – lipid nanoparticle encapsulating mRNAs encoding *OX40L*, *IL-23* and *IL-36* + durvalumab	Solid tumors/ lymphoma	NCT03739931 (1); Recruiting
Solid lipid	mRNA-5671/V941 – lipid nanoparticle with mRNA encoding 4 different KRAS ^21^ mutations + pembrolizumab	Solid tumors	NCT03948763 (1); Recruiting
Solid lipid	DCR-MYC – *MYC* targeting siRNA in stable lipid particle	Solid tumors / lymphoma	NCT02110563 (1); Terminated
Solid lipid	DCR-MYC - *MYC* targeting siRNA in stable lipid particle	Hepatocellular carcinoma	NCT02314052 (1/2); Terminated
Solid lipid	TKM-080301 - *PLK1* ^22^ targeting siRNA in lipid nanoparticle	liver cancer	NCT01437007 (1); Completed
Solid lipid	TKM-080301 - *PLK1* targeting siRNA in lipid nanoparticle	adrenocortical cancer	NCT01262235 (1/2); Completed
Solid lipid	TKM-080301 - *PLK1* targeting siRNA in lipid nanoparticle	hepatocellular carcinoma	NCT02191878 (1/2); Completed; [252]
Solid lipid	mRNA-4157 - mRNA personalized cancer vaccine + Pembrolizumab	Solid tumors	NCT03313778 (1); Recruiting; [254]
Solid lipid	ALN-VSP02 – *VEGF-A ^23^* and *KSP ^24^* siRNA in lipid nanoparticle	Solid tumors	NCT01158079 (1); Completed
**Biological Nanoparticles**			
Exosomes	MSC-derived exosomes with *KRAS* G12D siRNA	Pancreatic adenocarcinoma	NCT03608631 (1); Recruiting

^1^ AuNPs: gold nanoparticles; ^2^ BCL2L12: B-cell lymphoma 2 (BCL-2) like protein 12; ^3^ DOTAP: 1,2-Dioleoyloxy-3-trimethylammonium propane; ^4^ FUS1: nuclear fusion protein FUS1; ^5^ NSCLC: Non-small cell lung cancer; ^6^ DOTMA: 1,2-di-O-octadecenyl-3-trimethylammonium propane; ^7^ IL-2: interleukin-2; ^8^ cRAF: protein kinase RAF1; ^9^ EPH2: ephrin type-A receptor 2 precursor; ^10^ siRNA: small interfering RNA; ^11^ DOPC: 1,2-dioleoyl-sn-glycero-3-phosphatidylcholine; ^12^ TUSC2: tumor suppressor candidate 2; ^13^ mRNA: messenger RNA; ^14^ PKN3: protein kinase 3; ^15^ shRNA: short hairpin RNA; ^16^ STMN1: human stathmin 1; ^17^ saRNA: small/short activating RNA; ^18^ CEBPA: CCAAT enhancer-binding protein alpha; ^19^ GRB2: growth factor receptor bound protein 2; ^20^ OX40L: OX40 ligand (CD252); ^21^ KRAS: Kirsten rat sarcoma viral oncogene homolog; ^22^ PLK1: Polo-like kinase 1.; ^23^ VEGF-A: Vascular endothelial growth factor A; ^24^ KSP: kinesin spindle protein.

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
