# Peer review of "Gene Therapy in Cancer Treatment: Why Go Nano?"

_pharmaceutics, 2020, doi:10.3390/pharmaceutics12030233_

Round 1

Reviewer 1 Report

Summary

This is a comprehensive and well-researched review which nicely covers the field of nonviral gene therapy with a focus on cancer. The list of references is particularly valuable for researchers, and is admirably up-to-date. My main criticism is the rather uncritical tone of the review (see below), but this is a matter of style rather than a serious flaw.

General comments

1.1 There is only a brief section (lines 70-76) comparing viral with non-viral gene therapy, and this exists mainly to point out the shortcomings of viral vectors. A naive reader might imagine that viral vectors are "old news", with non-viral vectors destined to replace them in the near future. This is manifestly not true, as preclinical and clinical gene therapy effort is increasingly concentrated on viral vectors. To balance this, the authors should add to this section a few sentences admitting to the relative shortcomings of non-viral vectors: e.g., much lower transfection efficiency, and (usually) lower cell-targeting specificity (though EPR should be introduced here as an advantage in reference to cancer).

1.2 Another criticism of the non-viral gene therapy field in general (not just cancer) is that too much work seems to be confined to in vitro studies. Many of the excellent Tables very helpfully include an Application column which proves this point. The truth is that for non-viral gene therapy formulations, in vitro studies usually indicate efficacies that are not reproduced in the much more challenging in vivo environment. I would therefore like the authors (if they agree) to include in section 5 (say between "mechanisms." and "Due" (line 754)), a sentence like: "However, while in vitro proof-of-concept studies are a useful prerequisite, early progression to in vivo testing of novel nanomedicine approaches is to be encouraged as a means of rejecting formulations that have little prospect of clinical utility".

1.3 Figure 1 contrasts in vivo with ex vivo gene therapy. Ex vivo gene therapy is mainly discussed in lines 48-51. Do the authors agree with my impression that ex vivo gene therapy for _cancer_ gene therapy is a rather specialised subfield, with only a few specific applications? If so, please insert a sentence in this section saying something like "the bulk of cancer gene therapy research is concentrated on in vivo delivery approaches".

Specific comments

2.1 Section 2.3. microRNA targeted therapy. I am not sure how much miRNA needs to be delivered in the various approaches cited. It seems to me that quite large amounts might be needed if delivered systemically. Can the authors comment on this issue please?

2.2 Line 272: "chapter" -> "section"

2.3 Line 292 "gene" -> "gene therapy"

2.4 Section 2.7.3 CAR-T. This is the most important advance in cancer gene therapy by far and should be the first topic in this section! Please swap 2.7.1 with 2.7.3 (or similar) to achieve this. Also, please replace "based therapies" by "based viral gene therapies" on Line 333. It would be good in addition to explain how CAR-T based non-viral gene therapies might be better & safer than Kymriah et al, and what hurdles need to be overcome.

2.5 Table 1 Page 11. Advantages and disadvantages of nanoparticles used for gene delivery. I can't see why exosomes are included as "Biological nanoparticles" and viruses are not. You should at least acknowledge their existence -- for example, by adding a footnote. Change "Biological nanoparticles" to "Biological nanoparticles*", and in the footnote say "*Viruses are excluded as being outside the scope of this review"."

Author Response

Q1: There is only a brief section (lines 70-76) comparing viral with non-viral gene therapy, and this exists mainly to point out the shortcomings of viral vectors. A naive reader might imagine that viral vectors are "old news", with non-viral vectors destined to replace them in the near future. This is manifestly not true, as preclinical and clinical gene therapy effort is increasingly concentrated on viral vectors. To balance this, the authors should add to this section a few sentences admitting to the relative shortcomings of non-viral vectors: e.g., much lower transfection efficiency, and (usually) lower cell-targeting specificity (though EPR should be introduced here as an advantage in reference to cancer).

R1: We thank the Reviewer valuable comment that directed us to add in this section information regarding the shortcomings of the use of nanoparticles in gene therapy (lines 95-98).

Q2: Another criticism of the non-viral gene therapy field in general (not just cancer) is that too much work seems to be confined to in vitro studies. Many of the excellent Tables very helpfully include an Application column which proves this point. The truth is that for non-viral gene therapy formulations, in vitro studies usually indicate efficacies that are not reproduced in the much more challenging in vivo environment. I would therefore like the authors (if they agree) to include in section 5 (say between "mechanisms." and "Due" (line 754)), a sentence like: "However, while in vitro proof-of-concept studies are a useful prerequisite, early progression to in vivo testing of novel nanomedicine approaches is to be encouraged as a means of rejecting formulations that have little prospect of clinical utility".

R2: We added the sentence as suggested (lines 793-795).

Q3: Figure 1 contrasts in vivo with ex vivo gene therapy. Ex vivo gene therapy is mainly discussed in lines 48-51. Do the authors agree with my impression that ex vivo gene therapy for cancer gene therapy is a rather specialised subfield, with only a few specific applications? If so, please insert a sentence in this section saying something like "the bulk of cancer gene therapy research is concentrated on in vivo delivery approaches".

R3: The authors agree that ex vivo gene therapy for cancer is only directed at specific applications – this was also pointed out by Reviewer #3. As such, a new paragraph was added detailing the important development of ex vivo approaches for cancer therapeutics, with particular focus on CAR-T cells. Lines 331-337.

Q4: Section 2.3. microRNA targeted therapy. I am not sure how much miRNA needs to be delivered in the various approaches cited. It seems to me that quite large amounts might be needed if delivered systemically. Can the authors comment on this issue please?

R4: Although there are no miRNA-based therapies currently in the clinics, several phase 1 and phase 2 clinical trials have provided some insights into the performance of this approach. For example, according to Hanna et al (2019, doi: 10.3389/fgene.2019.00478), miRNAs based systemic delivery methods were shown to have similarities with injection and intravenous administrations. As explained in the manuscript (section 2.3), miRNA-based therapies are confronted by off-target effects and eventual miRNA-mediated toxicity. Hence, the passive or active targeting of miRNA, or the administration of intratumoral injections, will allow the accumulation of miRNA at the tumor, improving the efficacy of the therapy with lower outside effects.

Q5: Line 272: "chapter" -> "section"

R5: Edited as indicated (line 279).

Q6: Line 292 "gene" -> "gene therapy"

R6: Edited as indicated (line 299).

Q7: Section 2.7.3 CAR-T. This is the most important advance in cancer gene therapy by far and should be the first topic in this section! Please swap 2.7.1 with 2.7.3 (or similar) to achieve this. Also, please replace "based therapies" by "based viral gene therapies" on Line 333. It would be good in addition to explain how CAR-T based non-viral gene therapies might be better & safer than Kymriah et al, and what hurdles need to be overcome.

R7: The order of sections was altered so as to accommodate an introduction to “tumor vaccines” (now section 2.7.2) prior to description of the CAR-T cell therapy (now Section 2.7.3).

Also, to accommodate a suggestion by Reviewer #2 and #3, additional information was added to highlight the relevance of the CAR-T cell therapy (introduction of section 2. Gene therapy focused on cancer). Moreover, as per suggestion, a new paragraph was introduced at the end of section “2.7.2. CAR-T cells therapy” explaining why CAR-T cells from non-viral gene therapies need to be developed.

Q8: Table 1 Page 11. Advantages and disadvantages of nanoparticles used for gene delivery. I can't see why exosomes are included as "Biological nanoparticles" and viruses are not. You should at least acknowledge their existence -- for example, by adding a footnote. Change "Biological nanoparticles" to "Biological nanoparticles*", and in the footnote say "*Viruses are excluded as being outside the scope of this review"."

R8: As suggested, a footnote was added to table 1: “* Viruses were not included since they are out of the scope of the review”.

Reviewer 2 Report

The manuscript of Roma-Rodrigues et al intend to review the latest development of nanotechnology for its application in cancer treatment. Although the paper is informative, is trying to cover too many areas and is not well structured. It is therefore very difficult to find the key point of why nanomedicine could improve cancer treatment.

Mayor points:

  • There are too many chapters of the review without a focus in Cancer. Although it is always fundamental to introduce the basic of the different fields, in this review, the authors included too much information on general aspects of gene therapy and, overall of nanotechnology. Importantly, this excess of information misleads the reader from the focus of the review: the advantages and disadvantages of nanotechnology for the treatment of Cancer. In addition, there are too few objective comparisons of efficacy and safety of nanomedicine versus other viral and non-viral methods for the treatment of cancer. Crucial information on how nanomedicine is improving (based on real data from clinical trials) and/or could improve (based on data on preclinical models) is lacking and/or is difficult to find.    

  • Lines 60-64 (pg 2). This sentence is misleading. It doesn´t cover the differences between in vitro and in vivo. In addition, it is not realistic, since ex vivo approaches have reached better clinical results than in vivo.  Moreover, it is not clear what the authors want to say with “the requirement of proliferative advantage of transfected cells”. Finally, the main strategies for cancer gene therapy do NOT MAINLY aim tackling tumor cell division. In fact the most successful are based in immune-gene-therapy that do not take into account cell division.

  • Figure 1 lack important aspects of in vivo versus ex vivo and some information is not correct. For example ex vivo approaches does not always require invasive procedures (i.e. CAR-T cells, DC vaccination, etc).

  • Line 87. An important aspect the author mentionned is the low immunogenicity and toxicity of nanoparticles compared to viral vectors. However, there are no references to the low toxicity or immunogenicity of nanomaparticles. In the reference cited [30], the authors to not analyze these aspects and only stated that “Carbon nanotubes are cytotoxic as there is lipid membrane peroxidation due to residual metal catalysts. Owing to this toxicity, CNTs are known to downregulate adhesive proteins and increase cell death (aspect ratio dependent), but can be nontoxic to primary immune cells when functionalized appropriately”.

  • Figure 2 is too simple. Neither, the endothelium nor the extracellular matrix barriers are represented.

  • Lines 171-176.  The initial sentence on cancer gene therapy does not even mention the most successful approach: immunotherapy-based gene therapy.

  • Figure 3 should highlight does strategies reaching the market and phase III clinical trials.

  • In points from 2.1 to 2.6 the authors only mention different approaches without taking into account real therapeutic benefits in clinical trials. There are too many studies for the different cancer gene therapy approaches. In order to be informative, in this review I would rather focus in those approaches getting into clinical trials or those providing pivotal evidences of their potential in preclinical studies. 

  • Lines 293-295. The immunization gene therapy does not only consists in the enhancement of the immune system efficacy towards TME cells, but also (mainly) to tumor cells.

  • Lines 311-323. Although, it is one of the most promising strategies for tumor vaccines, NGS-neo-epitopes, there are other strategies that have reach clinical trials that should be mentioned first.

  • Table 2, table 3, table 4 and table 5 should include tumor efficacy data. Preclinical data? Focus in those that give rise to the clinical trials mentioned in Table 6.

  • Table 6 should include information on whether there are published data or not. It also include the efficacy and safety of the clinical trials with published data.

  • Lane 750 (Conclusions). This sentence should be modified to: Nanomedicine has been investigating different ways to providing answers to the limitations faced by viral-based gene delivery. At the moment there are not solid data showing real clinical improvements of nanoparticles-based gene therapy compared to viral vectors.

Minor points:

  • Some references are not appropriately included. The author should try to include either, pivotal studies or reviews.   

  • IN abstract:  The authors should indicate in the last paragraph of the abstract that the review will focus on the use of nanoparticles for CANCER gene therapy.

  • Line 270-271. Rephrase, plasmids are also DNA

  • Lines 271-272. Rephrase, CRISPR/Cas9 are used for gene editing.

  • Line 440: specific to the b-catenin GENE (not protein)

Author Response

Q1: There are too many chapters of the review without a focus in Cancer. Although it is always fundamental to introduce the basic of the different fields, in this review, the authors included too much information on general aspects of gene therapy and, overall of nanotechnology. Importantly, this excess of information misleads the reader from the focus of the review: the advantages and disadvantages of nanotechnology for the treatment of Cancer. In addition, there are too few objective comparisons of efficacy and safety of nanomedicine versus other viral and non-viral methods for the treatment of cancer. Crucial information on how nanomedicine is improving (based on real data from clinical trials) and/or could improve (based on data on preclinical models) is lacking and/or is difficult to find.    

R1: The points raised by the Reviewer are of relevance in the context of gene therapy. In fact, one major challenge relates to the tremendous amount of information available regarding gene therapy as a whole but with very little and limited information about the efficacy of nanomedicines for this goal. Indeed, despite the many in vitro applications of nanomedicines, very few in vivo efficacy evaluations have been documented and even less clinical trials applications. What is more, not all these studies use the same criteria to evaluate the efficacy of the nanoformulations, rendering virtually impossible to critically compare non-viral vectors with viral vectors.

As such, as we indicate in the abstract (and throughout the discussions), we focused on discussing the current trends in the design of nanoparticles towards the improvement of gene therapy directed at cancer related conditions.

Q2: Lines 60-64 (pg 2). This sentence is misleading. It doesn´t cover the differences between in vitro and in vivo. In addition, it is not realistic, since ex vivo approaches have reached better clinical results than in vivo.  Moreover, it is not clear what the authors want to say with “the requirement of proliferative advantage of transfected cells”. Finally, the main strategies for cancer gene therapy do NOT MAINLY aim tackling tumor cell division. In fact the most successful are based in immune-gene-therapy that do not take into account cell division.

R2: In the referred section we focus on gene therapies directed to the gene(tic) modulation of tumor cells and/or tumor microenvironment components. Hence, the immune-gene therapies were not contemplated as target of these genetic modification, since is these are healthy cells of the patient. Nevertheless, a sentence was added at the end of this section to clarify the relevance of ex-vivo cancer therapeutics based on immune-gene-therapy (lines 64-66).

Q3: Figure 1 lack important aspects of in vivo versus ex vivo and some information is not correct. For example ex vivo approaches does not always require invasive procedures (i.e. CAR-T cells, DC vaccination, etc).

R3: As mentioned above, we focused on gene therapies directed at gene(tic) modulation of tumor cells and/or tumor microenvironment components. Such assumption was clarified in figure 1 caption: “Figure 1. Delivery strategies used for gene therapy directly targeting tumor cells or tumor microenvironment components, their major advantages (preceded by a green checkmark) and disadvantages (preceded by a red cross).”

Q4: Line 87. An important aspect the author mentioned is the low immunogenicity and toxicity of nanoparticles compared to viral vectors. However, there are no references to the low toxicity or immunogenicity of nanoparticles. In the reference cited [30], the authors to not analyze these aspects and only stated that “Carbon nanotubes are cytotoxic as there is lipid membrane peroxidation due to residual metal catalysts. Owing to this toxicity, CNTs are known to downregulate adhesive proteins and increase cell death (aspect ratio dependent), but can be nontoxic to primary immune cells when functionalized appropriately”.

R4: We thank the Reviewer attention to this important detail. Reference 30 was misplaced - (it should be reference 29. A very recent review on nanoparticles immunogenicity was added to complete the provided information: doi: 10.1039/c9bm01643k.

Q5: Figure 2 is too simple. Neither, the endothelium nor the extracellular matrix barriers are represented.

R5: Indeed, figure 2 is a simplified schematic to highlight the critical players. Still, the term “blood vessel” was replaced by “tumor capillary, which is more correct.

Q6: Lines 171-176.  The initial sentence on cancer gene therapy does not even mention the most successful approach: immunotherapy-based gene therapy.

R6: An additional sentence was added to address this issue - “The immunization gene therapies, particularly chimeric antigen receptor (CAR) in T cells (CAR-T cells) based therapies, stands-out as they represent the group of cancer gene therapies in clinical practice.” (Lines 178-180).

Q7: Figure 3 should highlight does strategies reaching the market and phase III clinical trials.

R7: Even though we are inclined to agree with the Reviewer’s comment, we believe that this would add more complexity to Figure 3. What is more, this matter is addressed in section 4, where we highlight current trends towards the clinics (including clinical trials). Indeed, this is an introductory section to gene therapies focused on cancer.

Q8: In points from 2.1 to 2.6 the authors only mention different approaches without taking into account real therapeutic benefits in clinical trials. There are too many studies for the different cancer gene therapy approaches. In order to be informative, in this review I would rather focus in those approaches getting into clinical trials or those providing pivotal evidences of their potential in preclinical studies. 

R8: As mentioned before, this section intends to introduce the thematic to multi-disciplinary readers and wider audience ranging from biomaterials to cancer. That is why, the major advantages and disadvantages of each approach were mentioned along the section, together with those approaches currently in clinical trials and/or in clinical practice.

Q9: Lines 293-295. The immunization gene therapy does not only consists in the enhancement of the immune system efficacy towards TME cells, but also (mainly) to tumor cells.

R9: The sentence was edited so as to accommodate the suggestion, and now reads: “The immunization gene therapy consists in the enhancement of the immune system efficacy towards TME cells, with major focus on tumor cells.” Line 301.

Q10: Lines 311-323. Although, it is one of the most promising strategies for tumor vaccines, NGS-neo-epitopes, there are other strategies that have reach clinical trials that should be mentioned first.

R10: The introduction to section “2.7.1 tumor vaccines” was altered as per indication.

Q11: Table 2, table 3, table 4 and table 5 should include tumor efficacy data. Preclinical data? Focus in those that give rise to the clinical trials mentioned in Table 6.

R11: Following the suggestion, a new column was added to tables 2-5 describing the implications of therapy for tumor eradication, and a synopsis of in vitro and in vivo results. Due to the introduction of this column and to avoid repetition, the column “Highlights” was altered to “Description” and column “applications” removed. Also, references 181, 227 and 230 were removed as they were not directly related with cancer therapy. Since it was already referred in table 4, reference 226 was removed from table 5.

Still, we would like to refer that several of these studies did not end up in clinical trials. Nevertheless, they are relevant as they constitute recent breakthroughs towards application of nanomedicine in cancer therapy – for example, improving drug delivery to anatomical locations with difficult access.

Q12: Table 6 should include information on whether there are published data or not. It also include the efficacy and safety of the clinical trials with published data.

R12: The references of the published clinical trials were introduced into the table. The efficacy and safety of the clinical trials with published data were included throughout section 4.1.

Q13: Lane 750 (Conclusions). This sentence should be modified to: Nanomedicine has been investigating different ways to providing answers to the limitations faced by viral-based gene delivery. At the moment there are not solid data showing real clinical improvements of nanoparticles-based gene therapy compared to viral vectors.

R13: The sentence (line 789) was altered as per suggestion.

Q14: Some references are not appropriately included. The author should try to include either, pivotal studies or reviews.   

R14: The references were carefully re-analysed to ensure that the most important and recent were cited.

Q15: IN abstract:  The authors should indicate in the last paragraph of the abstract that the review will focus on the use of nanoparticles for CANCER gene therapy.

R15: It was changed accordingly in line 22.

Q16: Line 270-271. Rephrase, plasmids are also DNA

R16: The sentence was rephrased. Line 277.

Q17: Lines 271-272. Rephrase, CRISPR/Cas9 are used for gene editing.

R17: The sentence now reads: “The challenges in the delivery of the CRISPR/Cas9 system are similar to those of other gene editing, strategies detailed in chapter section 2.2. “Tumor suppressor genes replacement”. Lines 278-280.

Q18: Line 440: specific to the b-catenin GENE (not protein)

R18: Corrected (line 462).

Reviewer 3 Report

In this review article, Roma-Rodriguez and co-workers provide an extensive and detailed overview of the state-of-the-art non-viral tools for gene therapy in cancer.  The authors provide detailed strategies that can be used to target cancer cells or the immune system both in vitro and in vivo.

            The topic is of great interest and the references are up to date. The paper is well written, only a very minor grammatical checking may be necessary. I could not find any major deficits in the manuscript.

Minor comments

In the CAR-T cell section, it is unclear how non-viral delivery may apply, since the technology requires long lasting expression of the CAR in the T cells, which is achieved through stable transduction. Are there examples of non-viral gene transfer strategies used to enhance CAR-T cell function? Endogenous cancer cell-released exosomes have been exploited as a source of tumor antigens for tumor vaccination, see Squadrito et al., Nature Methods 2018. This may be interesting to comment.

Author Response

We thank the Reviewer for the motivating words.  

Q1: In the CAR-T cell section, it is unclear how non-viral delivery may apply, since the technology requires long lasting expression of the CAR in the T cells, which is achieved through stable transduction. Are there examples of non-viral gene transfer strategies used to enhance CAR-T cell function?

R1: Since this was also an issue raised by Referee #1 and #2, new information was added to section “2.7.2. CAR-T cells therapy”. Lines 331-337.

Q2: Endogenous cancer cell-released exosomes have been exploited as a source of tumor antigens for tumor vaccination, see Squadrito et al., Nature Methods 2018. This may be interesting to comment. 

R2: We thank reviewer suggestion. This study is indeed very interesting and a phrase referring it was introduced at the end of section 2.10 Targeting tumor cells derived exosomes (lines 390-394).

Round 2

Reviewer 2 Report

The authors have improved some aspects of the manuscript but some other part are unsolved:

  • The authors have not reduced the information on general aspects of gene therapy and nanotechnology and have not included more clear (based on preclinical or clinical data) advantages and disadvantages of nanotechnology for the treatment of Cancer
  • Figure 2. I understand that the aim is to provide a simplified overview of the critical players, but while in the cell context, the authors mention - endocytosis, endosome escape and the nuclei, they do hihlight the barriers that nanoparticles must overcome to reach the tumor cell membrane.

Author Response

Q1: The authors have not reduced the information on general aspects of gene therapy and nanotechnology…

R1: Considering the previous comments of the other reviewers, we believe that the manuscript as it is covers a broader scientific community from nanomaterials to biological areas, as it reviews the main aspects of gene therapy and nanotechnology and their impact in the development of novel (nano)therapeutic strategies for cancer.

Q2: (…) and have not included more clear (based on preclinical or clinical data) advantages and disadvantages of nanotechnology for the treatment of Cancer

 R2: To further highlight the advantages and bottlenecks of the application of nanoparticles in the clinics a new paragraph was introduced at the end of chapter 4.1 (lines 759-769) and in the conclusions section (lines 812-816).

Q3: Figure 2. I understand that the aim is to provide a simplified overview of the critical players, but while in the cell context, the authors mention - endocytosis, endosome escape and the nuclei, they do hihlight the barriers that nanoparticles must overcome to reach the tumor cell membrane.

R3: Figure 2 and respective caption were altered to include the fate of nanoparticles after entering the tumor – the EPR effect and the necessity to travel into hypoxic areas of the tumor.

We thank the reviewer comments that improved our manuscript.